# Inhibition of Sema4D/PlexinB1 signaling alleviates vascular dysfunction in diabetic retinopathy

Jie-hong Wu[1,†], Ya-nan Li[1,†], An-qi Chen[1,†], Can-dong Hong[1], Chun-lin Zhang[1], Hai-ling Wang[1], Yi-fan Zhou[1], Peng-Cheng Li[2], Yong Wang[3], Ling Mao[1], Yuan-peng Xia[1], Quan-wei He[1], Hui-juan Jin[1], Zhen-yu Yue[4] & Bo Hu[1,*] iD

## Abstract

Diabetic retinopathy (DR) is a common complication of diabetes and leads to blindness. Anti-VEGF is a primary treatment for DR. Its therapeutic effect is limited in non- or poor responders despite frequent injections. By performing a comprehensive analysis of the semaphorins family, we identified the increased expression of Sema4D during oxygen-induced retinopathy (OIR) and streptozotocin (STZ)-induced retinopathy. The levels of soluble Sema4D (sSema4D) were significantly increased in the aqueous fluid of DR patients and correlated negatively with the success of anti-VEGF therapy during clinical follow-up. We found that Sema4D/PlexinB1 induced endothelial cell dysfunction via mDIA1, which was mediated through Src-dependent VE-cadherin dysfunction. Furthermore, genetic disruption of Sema4D/PlexinB1 or intravitreal injection of anti-Sema4D antibody reduced pericyte loss and vascular leakage in STZ model as well as alleviated neovascularization in OIR model. Moreover, anti-Sema4D had a therapeutic advantage over anti-VEGF on pericyte dysfunction. Anti-Sema4D and anti-VEGF also conferred a synergistic therapeutic effect in two DR models. Thus, this study indicates an alternative therapeutic strategy with anti-Sema4D to complement or improve the current treatment of DR.

**Keywords** diabetic retinopathy; mDIA1; N-cadherin; Sema4D
**Subject Categories** Vascular Biology & Angiogenesis

## Introduction

Diabetic retinopathy (DR) is a common complication of diabetes and a leading cause of blindness in working-aged people (Cheung *et al*, 2010). The pathology of DR is characterized by early pericyte loss, vascular leakage, the formation of acellular capillaries, and late stage retinal ischemia, leading to an over-compensatory retinal neovascularization (Durham & Herman, 2011). Endothelial cell dysfunction is an important process in DR which leads to retinal vascular leakage and neovascularization. Due to the significant role played by vascular endothelial growth factor A (VEGF) in endothelial cell dysfunction, intravitreal injection (IVI) of its neutralizing antibody has been widely used to treat diabetic macular edema (DME) (Nguyen *et al*, 2009; Diabetic Retinopathy Clinical Research *et al*, 2015) and proliferative diabetic retinopathy (PDR) (Sivaprasad *et al*, 2017) in clinical practice. Although anti-VEGF therapy (such as aflibercept, ranibizumab) has provided some benefits, there are concerns surrounding anti-VEGF treatment. For many of the patients with DME who respond to anti-VEGF therapy, the response may only be temporary. These patients require frequent injections to control edema (Do *et al*, 2013; Channa *et al*, 2014; Ashraf *et al*, 2016). Furthermore, some patients are resistant to anti-VEGF therapy, presenting a challenge with regard to effective disease management (Channa *et al*, 2014; Ashraf *et al*, 2016). These patients may experience persistent macular edema over time despite frequent anti-VEGF injections alone (Channa *et al*, 2014; Ashraf *et al*, 2016). Moreover, the cost of anti-VEGF agents worldwide is huge, partly due to frequent injections; for example, the total drug expenditures on aflibercept in the United States alone was $2.2 billion in 2016 (Medicare, 2016). Therefore, there is a high demand for additional or alternative treatments to help those non- or poor responders to anti-VEGF and potentially reduce the number of injections.

In addition to endothelial cell dysfunction, pericyte loss is another hallmark of DR (Hammes *et al*, 2002). Pericytes are the vascular cells affected early in DR and pericyte loss leads to secondary changes in endothelial cells (Ejaz *et al*, 2008), which then initiate or aggravate several pathologic features including abnormal leakage, edema, acellular capillaries and ischemia, subsequently provoke proliferative neovascularization in the retina (Hammes *et al*, 2002; Ejaz *et al*, 2008; Durham & Herman, 2011). The development of new therapeutic agents that simultaneously target

1  Department of Neurology, Union Hospital, Tongji Medical College, Huazhong University of Science and Technology, Wuhan, China
2  Department of Ophthalmology, Union Hospital, Tongji Medical College, Huazhong University of Science and Technology, Wuhan, China
3  Aier School of Ophthalmology, Wuhan Aier Eye Hospital, Central South University, Wuhan, China
4  Department of Neurology and Department of Neuroscience, The Friedman Brain Institute, Icahn School of Medicine at Mount Sinai, New York, NY, USA
   *Corresponding author. Tel: +86 27 85726028; Fax: +86 27 85726028; E-mail: hubo@mail.hust.edu.cn
   †These authors contributed equally to this work as first authors

endothelial cells and pericytes may not only control the disease progression, but could also improve therapeutic outcomes.

Semaphorins and their receptors (Plexins) were initially discovered as axon guidance cues in development (Roth *et al*, 2009; Worzfeld & Offermanns, 2014). They have also been shown to play important roles in a variety of vascular pathophysiological processes (Roth *et al*, 2009; Worzfeld & Offermanns, 2014). To understand the role of semaphorin molecules in the pathogenesis of DR, we performed a systematic analysis of the expression of semaphorins in oxygen-induced retinopathy (OIR) and streptozotocin (STZ)-induced retinopathy models. We found that Sema4D was significantly increased in both models as well as in the aqueous fluid of DR patients, and the elevated aqueous fluid levels of Sema4D were associated with a poor response to anti-VEGF therapy. We showed that Sema4D induced endothelial cell dysfunction and triggered pericyte loss in DR. Moreover, our study not only revealed the therapeutic advantage of anti-Sema4D over anti-VEGF during multiple injections, but also suggested an improved therapy for DR with a combination of anti-VEGF treatment and the suppression of Sema4D/PlexinB1 signaling.

## Results

### Elevated Sema4D expression in mouse models and in the aqueous fluid of patients with diabetic retinopathy

To identify specific semaphorins that may contribute to the development of DR, we first performed a screening of the expression of semaphorins in two different models for DR: an STZ-induced mouse DR model, which develops only retinal vascular leakage without neovascularization (Olivares *et al*, 2017), and oxygen-induced retinopathy (OIR), an animal model of retinopathy of prematurity (ROP) with neovascularization and vascular leakage (Connor *et al*, 2009). While we found increased mRNA levels in multiple semaphorins, Sema4D was the most obvious semaphorin that increased in both models (Fig 1A and B).

Sema4D, a membrane-bound protein, can be shed from the cell surface via proteolytic cleavage to yield a soluble form (sSema4D) during different disease conditions (Maleki *et al*, 2016). To verify the potential involvement of Sema4D in human DR, we examined the protein levels of sSema4D in aqueous fluid from patients with DR, selected according to their macular thickness as determined by optical coherence tomography (OCT) and 3D retinal maps

(Fig 1C). Enzyme-linked immunosorbent assays (ELISA) revealed that sSema4D levels were robustly increased in the aqueous fluid of DR patients compared to that of the control patients (Fig 1D). Moreover, we found that the levels of sSema4D negatively correlated with changes in the central subfield thickness (CST) and macular volume (MV) in response to anti-VEGF therapy (Fig 1E and F).

### Sema4D is increased in retinas of oxygen-induced retinopathy and streptozotocin-induced diabetes

We assessed the retinal levels of Sema4D at different time points in the OIR model. Compared with age-matched room air controls, the mRNA and protein levels of Sema4D increased as a function of time during the neovascularization phase of OIR (Fig 2A–C). In STZ-treated mice, Sema4D protein levels were significantly increased at 3 and 6 months, compared to those in age-matched control mice (Fig 2D–G).

We then determined the cell type that expressed Sema4D. Immunofluorescence staining showed that the increased Sema4D signals mainly localized in GFAP$^+$ glial cells (astrocytes and Müller cells) in the OIR model (Fig 2H). Fluorescence *in situ* hybridization also indicated that Sema4D localized in GFAP$^+$ glial cells in OIR retinas (Fig 2I). Since retinal hypoxia contributes to both DME and PDR (Campochiaro *et al*, 2016), we used hypoxia stimulation to partly mimic the disease condition. By using isolated primary glial cells, we found that the mRNA levels of Sema4D were evidently increased when the glial cells were exposed to hypoxia (Fig 2J). Meanwhile, the secreted protein levels of Sema4D (sSema4D) from a conditional medium were also increased after hypoxia (Fig 2K).

### Sema4D is transcriptionally up-regulated by IRF1

To identify the mechanism that regulates the increase of Sema4D, we performed a bioinformatic search for the potential transcription factors that may regulate its expression. By using JASPAR software (http://jaspar.genereg.net), we predicted a transcription factor, IRF1, as a candidate. We first detected the expression of IRF1 in glial cells and found the protein levels of IRF1 increased over time after hypoxia stimulation (Appendix Fig S1A and B). To assess whether IRF1 is required for Sema4D expression, we utilized siRNA to silence IRF1 expression (Appendix Fig S1C). We found that IRF1 siRNA treatment blunted the induction of Sema4D mRNA upon hypoxia stimulation (Appendix Fig S1D). Furthermore, chromatin

---

**Figure 1. Elevated Sema4D expression in mouse models and in the aqueous fluid of patients with DR.**

A   The mRNA levels of semaphorins in age-matched controls in room air or OIR retinas at P17 were analyzed by qPCR (*n* = 6 per group, \*P < 0.05 compared with room air retinas).

B   The mRNA levels of semaphorins in retinas 6 months after DM onset were analyzed by qPCR (*n* = 6 per group, \*P < 0.05 compared with vehicle group).

C   Optical coherence tomography (OCT) and 3D retinal maps from control patients versus DME patients. The purple and light blue lines in the lower panels indicate the scanning level of OCT on the retinas.

D   The levels of sSEMA4D in the aqueous fluid of DME and control individuals were measured at baseline before initial anti-VEGF therapy (*P* < 0.05 compared with control patients).

E, F   Central subfield thickness (CST) and macular volume (MV) were measured by OCT at baseline before initial anti-VEGF therapy and 6 months after initial anti-VEGF therapy. The CST and MV changes were calculated as the baseline value minus the value at 3 months. The correlation curves between the levels of sSema4D in aqueous fluid with changes of CST and MV in response to anti-VEGF therapy are shown.

Data information: Data are means ± SD. Statistical test and *P*-values are reported in Appendix Table S3.

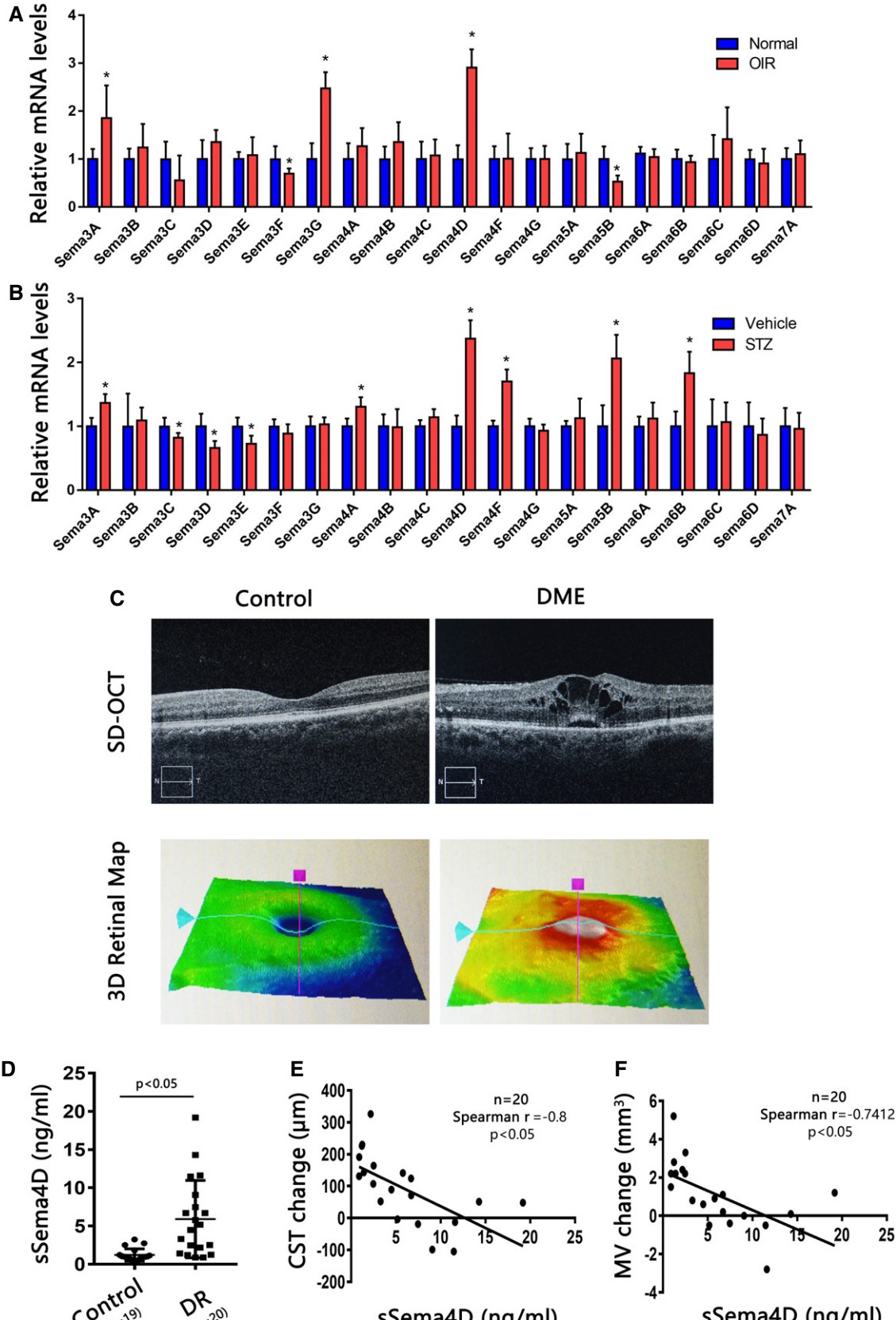

**Figure 1.**

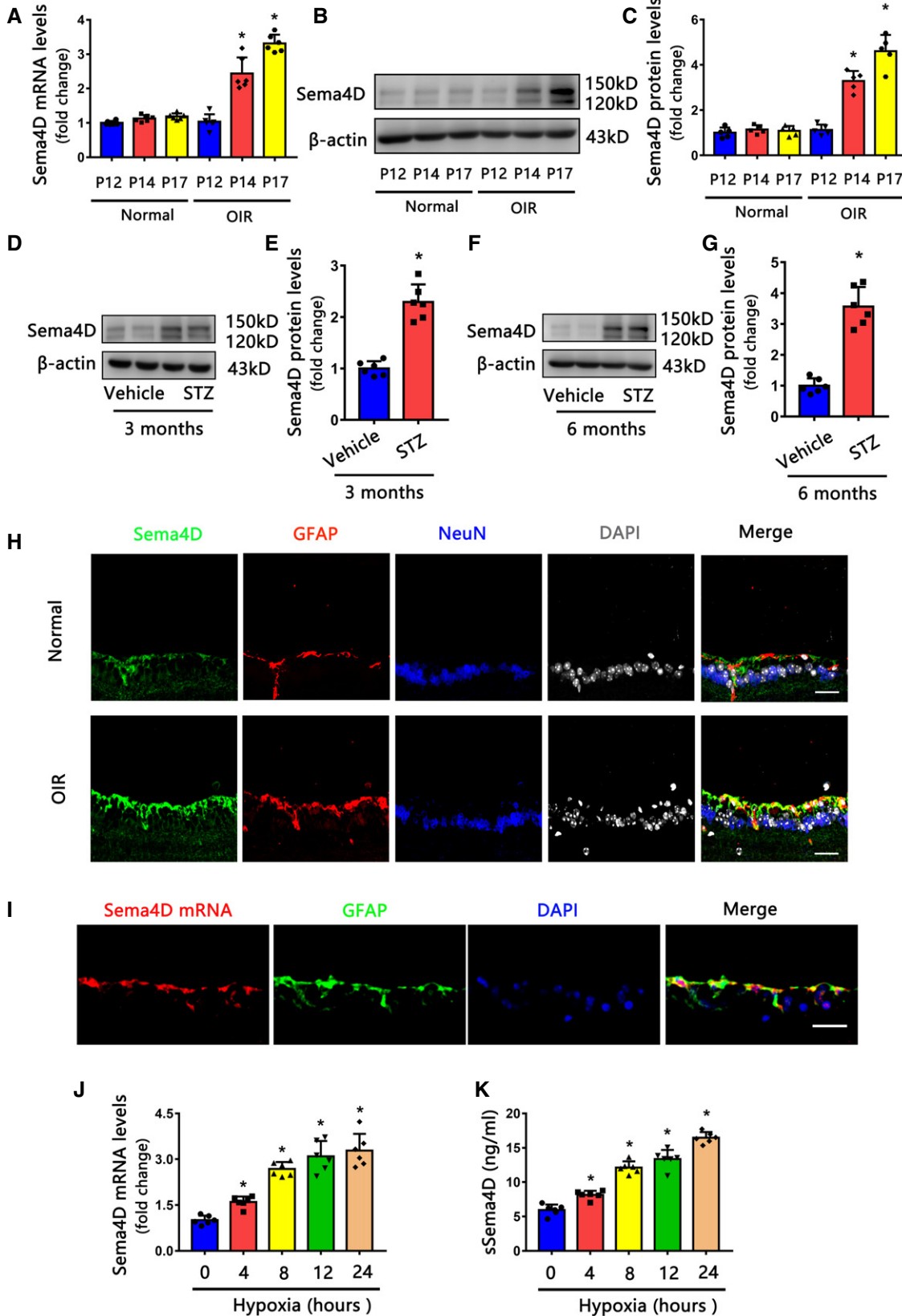

**Figure 2.**

**Figure 2. Glial Sema4D is increased in DR models.**

A  Sema4D mRNA levels in OIR retinas compared with age-matched controls in room air retinas ($n$ = 6, *$P$ < 0.05 compared with room air retinas).

B, C  Western blots (B) and quantification (C) of Sema4D protein levels in OIR retinas compared with age-matched controls in room air retinas ($n$ = 5, *$P$ < 0.05 compared with room air retinas).

D–G  Western blots (D, F) and quantification (E, G) of Sema4D protein levels in STZ retinas at three and 6 months of diabetes ($n$ = 6 per group, *$P$ < 0.05 compared with vehicle group).

H  Immunofluorescence staining of Sema4D (green) in OIR retinas with GFAP (red), NeuN (blue), and DAPI (gray) ($n$ = 5), and bars indicate 25 μm.

I  Fluorescence *in situ* hybridization with Cy3-labeled RNA probes targeting Sema4D followed by immunofluorescence staining with GFAP (green) in OIR retinas ($n$ = 5), and scale bars indicate 25 μm.

J, K  The mRNA levels of Sema4D in cell lysates (J) and secreted protein levels of Sema4D (sSema4D) from conditional medium (K) were increased in primary glial cells after hypoxia ($n$ = 6 per group, *$P$ < 0.05 compared with 0 h group).

Data information: Data are means ± SD. Statistical test and *P*-values are reported in Appendix Table S3.
Source data are available online for this figure.

immunoprecipitation (ChIP) assays demonstrated that hypoxia stimulation induced the recruitment of IRF1 transcription factor to the putative binding site of the Sema4D promoter (Appendix Fig S1E and F). Indeed, IRF1 was increased in retinas during OIR and STZ-induced diabetes, which paralleled the expression pattern change of Sema4D (Appendix Fig S1G–J).

### ADAM17 induces Sema4D shedding in glial cells

Previous studies have reported that Sema4D, a membrane-bound protein, is shed from the cell surface via proteolytic cleavage by certain matrix metalloproteinases (ADAM10, MMP14, ADAMTS4, ADAM17) to yield a soluble form (Maleki *et al*, 2016). To determine the specific metalloproteinases involved in Sema4D shedding in glial cells, we utilized siRNA to silence individual metalloproteinases (Appendix Fig S2A). We found that silencing ADAM17 only caused a significant decrease of sSema4D in the conditional medium of glial cells (Appendix Fig S2B). Moreover, the treatment of glial cells with TAPI-1, an ADAM17 inhibitor, markedly reduced hypoxia-stimulated sSema4D concentration in the supernatants of the medium of the glial cells in a dose-dependent manner (Appendix Fig S2C). Indeed, ADAM17 protein levels were up-regulated in both OIR and STZ retinas (Appendix Fig S2D–G).

### Sema4D knockout attenuates pathologic retinal neovascularization and vascular leakage

To determine the role of Sema4D in pathologic retinal neovascularization and vascular leakage, we generated a genetic knockout (KO) of Sema4D mice (Appendix Fig S3A–C). By using isolectin B4 staining and Evans blue leakage assays, we found that Sema4D-KO mice showed an obvious reduction of pathologic retinal neovascularization as well as vascular leakage at P17 compared with littermate wild-type (WT) controls in the OIR model (Fig 3A–C). Consistently, the average number of pre-retinal neovascular cells in the retinas of Sema4D-KO mice was significantly decreased compared with littermate WT controls (Fig 3D and E).

To evaluate whether Sema4D KO inhibits the process of DR in an STZ model, we induced diabetes in Sema4D-KO mice and age-matched WT mice. Evans blue leakage assays indicated that diabetic Sema4D-KO mice showed lower vascular leakage than diabetic WT mice (Fig 3F and G). Furthermore, Sema4D KO alleviated diabetes mellitus-induced acellular capillary formation (Figs 3H and I, and EV1A and B). Moreover, by staining pericytes with PDGFRβ and desmin markers, we found that diabetes-induced pericyte loss was markedly blocked by Sema4D KO (Figs 3J–L and EV1C and D).

In addition, we evaluated the effect of Sema4D KO on normal retinal vessel development. There was no significant difference between Sema4D-KO mice and littermate WT mice in relation to retinal vascular development at P5, as analyzed by the vascular area, vascular outgrowth, vascular branch points, and sprout number (Appendix Fig S3D–H).

### Sema4D promotes endothelial cell migration and permeability through the PlexinB1 receptor *in vitro*

We next investigated the function of Sema4D in endothelial cells *in vitro*. Wound-healing assays indicated that recombinant Sema4D promoted endothelial cell migration in a dose-dependent manner (Fig EV2A and B). To assess the effect of Sema4D on vascular permeability, trans-endothelial electrical resistance (TEER) and permeability to dextran were measured on endothelial monolayers. We found that recombinant Sema4D decreased the TEER value of endothelial monolayers and increased their permeability to dextran (Fig EV2C and D).

Previous studies have indicated that Sema4D binds with a high affinity to receptor PlexinB1, which mediates the Sema4D signaling afferent (Matsunaga *et al*, 2016). To evaluate whether PlexinB1 mediates the effects in the above condition, we used lentivirus-mediated CRISPR/Cas9 gene disruption of PlexinB1 (CRISPR-plB1) in endothelial cells (Fig EV2E) and found that the knockdown of PlexinB1 expression significantly blocked the stimulation effect of Sema4D on endothelial cell migration (Fig EV2F and G). Meanwhile, PlexinB1 knockdown also substantially abolished the Sema4D-elicited increased permeability of endothelial monolayers (Fig EV2H and I).

### Sema4D/PlexinB1 regulates endothelial cell function via mDIA1-Src signaling

Through the mining of a large-scale protein–protein network database, we predicted that C210RF59, CENPO, CENPU, mDIA1 (gene name: DIAPH1), HNRNPD, and SGTB were potential binding partners of PlexinB1 (Hein *et al*, 2015). Among these proteins, only mDIA1 has been reported to participate in angiogenesis and vascular permeability (Gavard *et al*, 2008; Zhou *et al*, 2014b, 2018a). In addition, mDIA1 has been shown to form a complex with Src to regulate Src tyrosine kinase signaling (Tominaga *et al*, 2000; Gavard *et al*, 2008; Zhou *et al*, 2018a). Furthermore, Src tyrosine kinase could phosphorylate focal adhesion kinase (Fak) and vascular

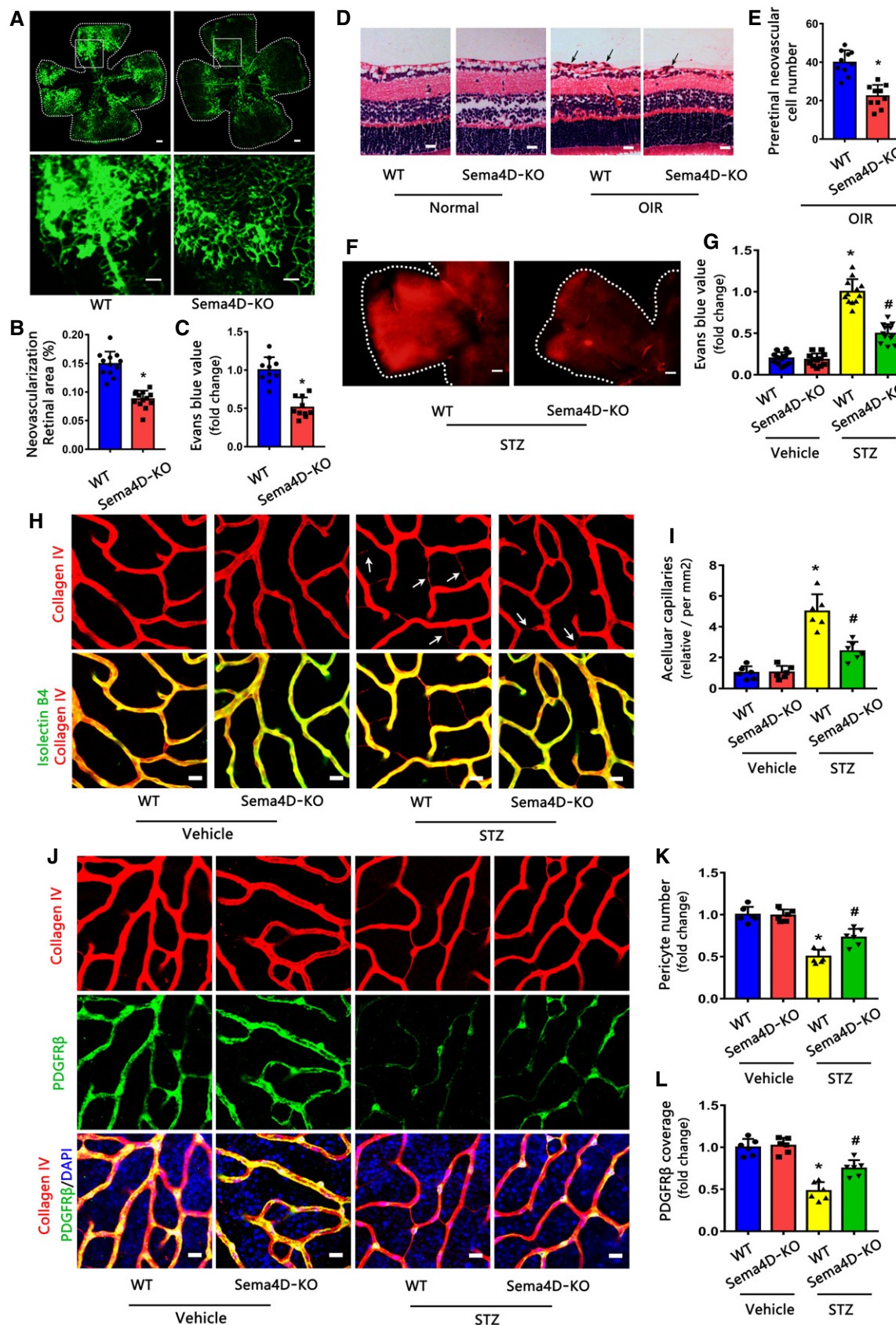

Figure 3.

**Figure 3. Sema4D knockout attenuates pathological retinal neovascularization and vascular leakage.**

A–C Isolectin B4 staining and Evans blue assays were performed to examine pathological retinal neovascularization and vascular leakage in whole-mount retinas at P17 in the OIR model with or without Sema4D. For isolectin B4 staining, multiple overlapping (10–20% overlap) images were obtained with a 4× lens on a fluorescence microscope. The images were merged to visualize the entire retinas. The upper images are representative composite images of entire retinas. The lower images are a cropped enlarged view from the upper entire retina images. The processes were used in all the following isolectin B4 staining figures for the entire retina images ($n = 12$ for WT in B, $n = 13$ for Sema4D-KO in B, $n = 10$ in C, scale bars indicate 200 μm for upper pictures and 100 μm for lower pictures, $*P < 0.05$ compared with WT group).

D, E Representative HE staining images and quantification of pre-retinal neovascular cells in retinal cross-sections at P17 in the OIR model with or without Sema4D ($n = 10$. Arrows indicate pre-retinal neovascular cells. Scale bars indicate 20 μm, $*P < 0.05$ compared with WT group).

F, G Evans blue assays were used to test vascular leakage at 3 months in the STZ model with or without Sema4D. Representative Evans blue fluorescent images are shown. The calculated extracted Evans blue values were used for quantification ($n = 12$, scale bars indicate 200 μm, $*P < 0.05$ compared with WT + vehicle group, $\#P < 0.05$ compared with WT + STZ group).

H, I Immunofluorescence staining of isolectin B4 (green) with collagen IV (red) showed that Sema4D knockout attenuated acellular capillary formation at 3 months in the STZ model ($n = 6$. Arrows indicate acellular capillaries. Scale bars indicate 10 μm, $*P < 0.05$ compared with WT + vehicle group, $\#P < 0.05$ compared with WT + STZ group).

J–L Immunofluorescence staining of PDGFRβ (green, a pericyte marker) with collagen IV (red) at 3 months in the STZ model ($n = 6$, scale bars indicate 10 μm, $*P < 0.05$ compared with WT + vehicle group, $\#P < 0.05$ compared with WT + STZ group).

Data information: Data are means ± SD. Statistical test and $P$-values are reported in Appendix Table S3.

endothelial cadherin (VE-cadherin), a main component of endothelial adherens junctions, to induce angiogenesis and vascular permeability (Dejana et al, 2008; Zhao & Guan, 2011; Miloudi et al, 2016). Therefore, we next questioned whether PlexinB1 could form a complex with mDIA1/Src upon Sema4D treatment, thereby activating Src tyrosine kinase signaling to regulate endothelial cell function. We performed co-immunoprecipitation to detect complex formation with the indicated antibodies (Fig 4A), and the results showed that the PlexinB1-m DIA1/Src complex formation increased upon stimulation with Sema4D. Meanwhile, we found that Sema4D induced the phosphorylation of Src, VE-cadherin, and Fak (Fig 4B and C) in a time-dependent manner. To determine whether Sema4D works through mDIA1, we knocked down mDIA1 expression using specific siRNA (Fig 4D). The reduced expression of mDIA1 blocked the Sema4D-induced phosphorylation of Src, Fak, and VE-cadherin (Fig 4E and F). Furthermore, silencing mDIA1 markedly inhibited Sema4D-induced endothelial cell migration and permeability (Fig 4G–J). It is known that the interaction between VE-cadherin and several partners, including p120-catenin and β-catenin, prevents VE-cadherin internalization and maintains the stabilization of endothelial adherens junctions in normal conditions (Dejana et al, 2008). Phosphorylation of VE-cadherin could weaken the interaction and promote VE-cadherin internalization thus triggering a discontinuity of VE-cadherin to increase vascular permeability in disease conditions (Dejana et al, 2008). Consistently, we also found that Sema4D promoted VE-cadherin internalization and the effect was prevented by mDIA1 silencing (Fig 4K and L).

To test whether Sema4D/PlexinB1 regulates Src-dependent function in endothelial cells, we used an inhibitor of Src (KX2-391). Pretreatment of the cells with KX2-391 markedly blocked the Sema4D-dependent induction of Fak and VE-cadherin phosphorylation, endothelial cell migration, and permeability (Appendix Fig S4A–E). Meanwhile, a selective FAK inhibitor (GSK2256098) also partly blocked Sema4D-induced endothelial cell migration and permeability (Appendix Fig S4F–H).

## Sema4D/PlexinB1 induces pericyte migration and N-cadherin internalization to worsen the vascular permeability

Since the above results indicated that Sema4D KO prevented diabetes-induced pericyte loss in retinas, we next explored the effect of Sema4D in pericytes. We used another permeability detection model by co-culturing pericytes with endothelial cells in the Transwell system in vitro. Consistently, Sema4D treatment caused a dose-dependent decrease in the TEER value and an increase in the permeability to dextran in the co-culture system (Fig 5A and B). Interestingly, we found that the Sema4D-induced fold changes of the TEER value and leaked dextran were more evident in the co-culture system than in previous sole endothelial monolayers (Fig EV2C and D). This result seems to indicate that Sema4D may have an additional role in pericytes. We used fluorescence-activated cell sorting (FACS) to isolate endothelial cells and pericytes in retinas and found that PlexinB1 mRNA was expressed in the two freshly sorted cell types (Fig 5C; Appendix Fig S5A). We also noted that the protein levels of PlexinB1 were higher in cultured pericytes than endothelial cells (Fig 5D and E; Appendix Fig S5B). We then knocked down PlexinB1 expression in pericytes and in endothelial cells, separately (Fig 5F and G) and found that the knockdown of PlexinB1 in pericytes reversed Sema4D-induced permeability to a greater extent than the knockdown of PlexinB1 in endothelial cells in the co-culture system (Fig 5H and I).

Previous studies have reported that pericyte apoptosis and migration are the two main underlying mechanisms in pericyte-associated pathologies in DR (Pfister et al, 2008; Geraldes et al, 2009; Park et al, 2014b; Kim et al, 2015; Hu et al, 2017). We found that Sema4D had little effect on pericyte apoptosis (Appendix Fig S5C and D), but significantly promoted pericyte migration in a dose-dependent manner (Appendix Fig S5E and F). The stimulatory effect was reversed by PlexinB1 knockdown (Fig 5J and K). N-cadherin forms complexes with p120-catenin and β-catenin in the pericyte membrane, mediating pericytes' adhesion to vessels (Navarro et al, 1998; Paik et al, 2004; Li et al, 2011; Hu et al, 2017). When phosphorylated by Src tyrosine kinase in disease conditions, the complex is dissociated, accompanied by the internalization of N-cadherin, leading to pericyte migration (Qi et al, 2006; Hu et al, 2017). We then examined whether N-cadherin dysfunction participates in Sema4D-induced pericyte migration. Our results revealed that Sema4D induced Src phosphorylation over time in pericytes (Fig 5L and M). Meanwhile, Sema4D caused the dissociation of N-cadherin from p120-catenin and β-catenin (Fig 5N) and

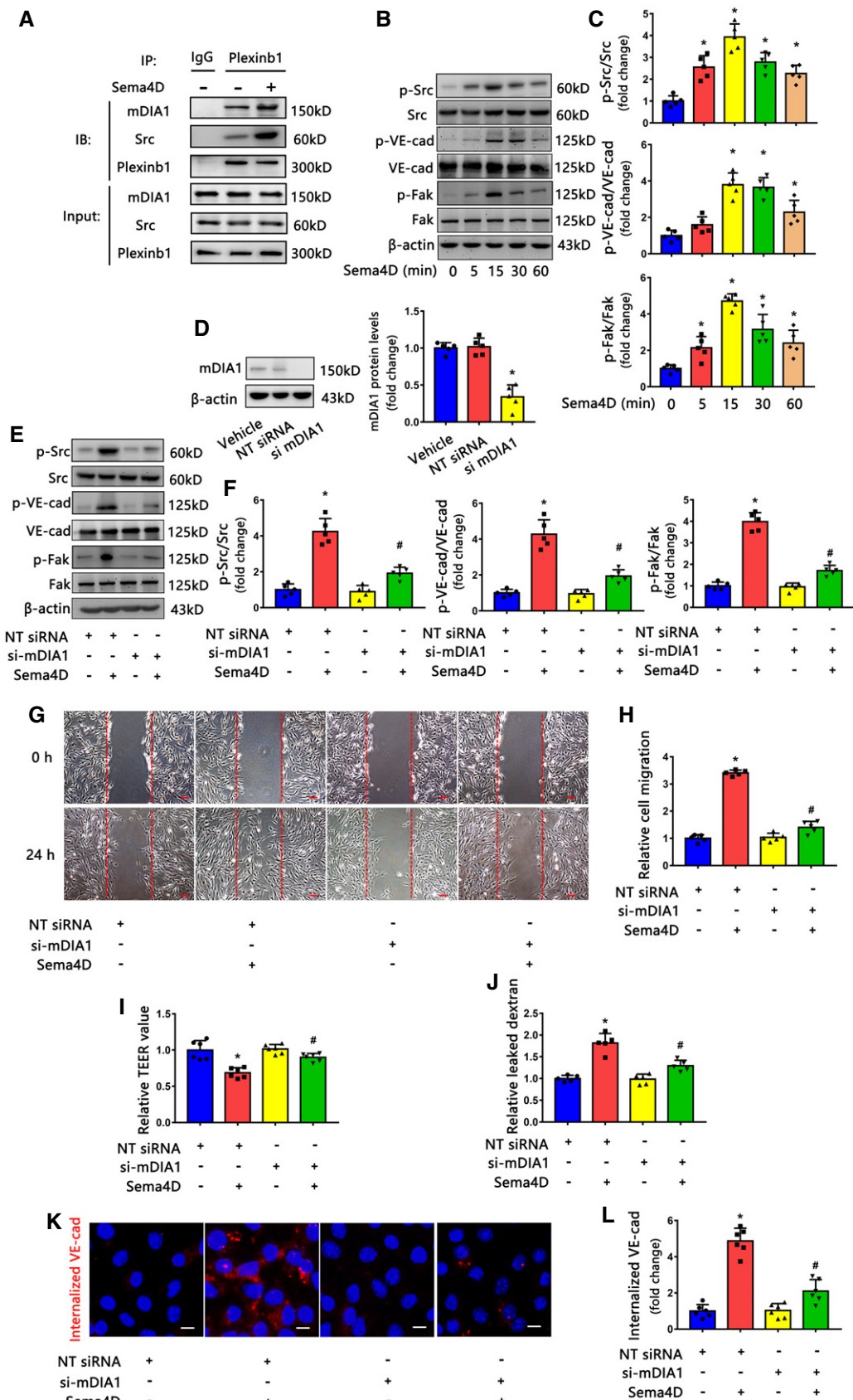

**Figure 4.**

Figure 4. Sema4D/PlexinB1 regulates endothelial cell function through mDIA1.

A    Endothelial cell lysates were prepared and used for an immunoprecipitation assay with anti-PlexinB1 antibody after the treatment with Sema4D for 30 min. Immunoblots (IB) were performed with antibodies against mDIA1 and Src ($n = 4$).

B, C    Western blotting showed that Sema4D induced the phosphorylation of Src, VE-cadherin, and Fak in endothelial cells in a time-dependent manner ($n = 5$, $*P < 0.05$ compared with 0 min group).

D    The expression of mDIA1 was silenced by mDIA1 siRNA in endothelial cells ($n = 5$, $*P < 0.05$ compared with NT siRNA group).

E–L    Endothelial cells transfected with NT siRNA or mDIA1 siRNA were treated with or without 1600 ng/ml recombinant Sema4D, followed by Western blotting for the levels of phosphorylation of Src, VE-cadherin, and Fak after 30-min treatment (E and F) with specific antibodies. Wound healing (G and H), TEER value (I), dextran permeability (J), and VE-cadherin internalization (K and L) were measured ($n = 5$ in F, H, J; $n = 6$ in I, L. The vertical red lines indicate the border of the wound in G. Scale bars indicate 100 μm for wound healing and 20 μm for VE-cadherin internalization, $*P < 0.05$ compared with NT siRNA group, $^{\#}P < 0.05$ compared with NT siRNA + Sema4D group).

Data information: Data are means ± SD. Statistical test and *P*-values are reported in Appendix Table S3.
Source data are available online for this figure.

promoted the internalization of N-cadherin (Fig 5O and P). Furthermore, treatment with Src inhibitor markedly reversed the Sema4D-induced N-cadherin/p120-catenin/β-catenin complex dissociation, N-cadherin internalization, and pericyte migration (Fig 5N–S).

### Knockdown of PlexinB1 alleviates pathologic retinal neovascularization and vascular leakage *in vivo*

We then verified the involvement of PlexinB1 *in vivo*. To knockdown PlexinB1 in endothelial cells *in vivo*, we used Tie2-Cre mice to specifically express Cre recombinase in endothelial cells. Adenoviruses containing a potent expression of PlexinB1 shRNA that was activated by Cre recombinase-mediated recombination were intravitreally injected into these mice (Fig 6A). The GFP staining showed that the shRNA was mainly expressed in endothelial cells in OIR retinas (Fig 6B). Western blotting confirmed that PlexinB1 was knocked down in retinas (Fig 6C and D). Our results indicated that the knockdown of PlexinB1 in endothelial cells reduced the pathologic retinal neovascularization and vascular leakage in the OIR model (Fig 6E–G). Meanwhile, we used PDGFRβ-Cre mice combined with the adenovirus to knockdown PlexinB1 in pericytes in the STZ model (Fig 6H–K). We found that the knockdown of PlexinB1 in endothelial cells or pericytes, respectively, alleviated vascular leakage in the STZ model (Fig 6L and M).

### Sema4D knockout attenuates the PlexinB1 downstream signaling pathways *in vivo*

We next verified the changes in PlexinB1 downstream signaling *in vivo* by using Sema4D-KO mice. We found that the phosphorylation of Src, VE-cadherin, and Fak in retinas was significantly increased in the OIR model compared with the control; however, Sema4D-KO mice showed decreased phosphorylation of Src, VE-cadherin, and Fak in their retinas compared with littermate WT controls in the OIR model (Fig EV3A and B).

Previous studies have indicated that VE-cadherin and N-cadherin internalization could lead to a discontinuity of VE-cadherin and a reduction of N-cadherin coverage on vessels (Dejana *et al*, 2008; Yamamoto *et al*, 2015; Hu *et al*, 2017). In accordance with the previous studies (Dejana *et al*, 2008; Hu *et al*, 2017), we also detected a discontinuity of VE-cadherin and a reduction of N-cadherin coverage in STZ-induced diabetic retinopathy compared with the control group (Fig EV3C–F). Moreover, we found that Sema4D KO could rescue VE-cadherin continuity and N-cadherin coverage of retinal vessels in the STZ model (Fig EV3C–F).

### Anti-Sema4D antibody shows a therapeutic advantage over anti-VEGF during multiple injections

Our above observation suggests that the aberrant increase of Sema4D induces endothelial cell dysfunction and triggers pericyte

Figure 5. Sema4D/PlexinB1 induced pericyte migration and N-cadherin internalization to worsen the vascular permeability.

A, B    The TEER value and dextran permeability were measured in a pericyte (PC) and endothelial cell (EC) co-culture model after recombinant Sema4D treatment ($n = 6$ in A, $n = 5$ in B, $*P < 0.05$ compared with 0 ng/ml Sema4D group).

C    The mRNA levels of PlexinB1 in FACS-sorted endothelial cells and pericytes in retinas ($n = 6$, $*P < 0.05$ compared with EC group).

D, E    Western blots (D) and quantification (E) of PlexinB1 protein levels in cultured pericytes and endothelial cells *in vitro* ($n = 6$, $*P < 0.05$ compared with EC group).

F, G    Western blots (F) and quantification (G) of PlexinB1 protein levels in the CRISPR-plB-mediated knockdown of PlexinB1 in pericytes ($n = 5$, $*P < 0.05$ compared with CRISPR-wt group).

H, I    Knockdown of PlexinB1 in pericytes reversed Sema4D-induced permeability to a greater extent than the knockdown of PlexinB1 in endothelial cells in the co-culture system ($n = 6$ in H, $n = 5$ in I, $^{\#}P < 0.05$ compared with CRISPR-wt + Sema4D group).

J, K    Transwell migration assays showed that the stimulatory effect of Sema4D on pericyte migration was blocked by PlexinB1 knockdown ($n = 6$, scale bars indicate 100 μm, $*P < 0.05$ compared with CRISPR-wt group, $^{\#}P < 0.05$ compared with CRISPR-wt + Sema4D group).

L, M    Western blotting showed that Sema4D induced the phosphorylation of Src in pericytes ($n = 5$, $*P < 0.05$ compared with 0 min group).

N–S    Pericytes pretreated with KX2-391, an inhibitor of Src, were stimulated with or without 1,600 ng/ml recombinant Sema4D. For N-cadherin internalization and immunoprecipitation assays, the cells were treated with recombinant Sema4D for 4 h. (N) Cell lysates were used for immunoprecipitation with anti-N-cadherin antibody and then immunoblotted with antibodies against β-catenin and p120-catenin ($n = 4$). Pericyte N-cadherin internalization (O and P) ($n = 6$, scale bars indicate 20 μm) and migration (R and S) ($n = 6$, scale bars indicate 100 μm) were evaluated ($*P < 0.05$ compared with control group, $^{\#}P < 0.05$ compared with Sema4D group).

Data information: Data are means ± SD. Statistical test and *P*-values are reported in Appendix Table S3.
Source data are available online for this figure.

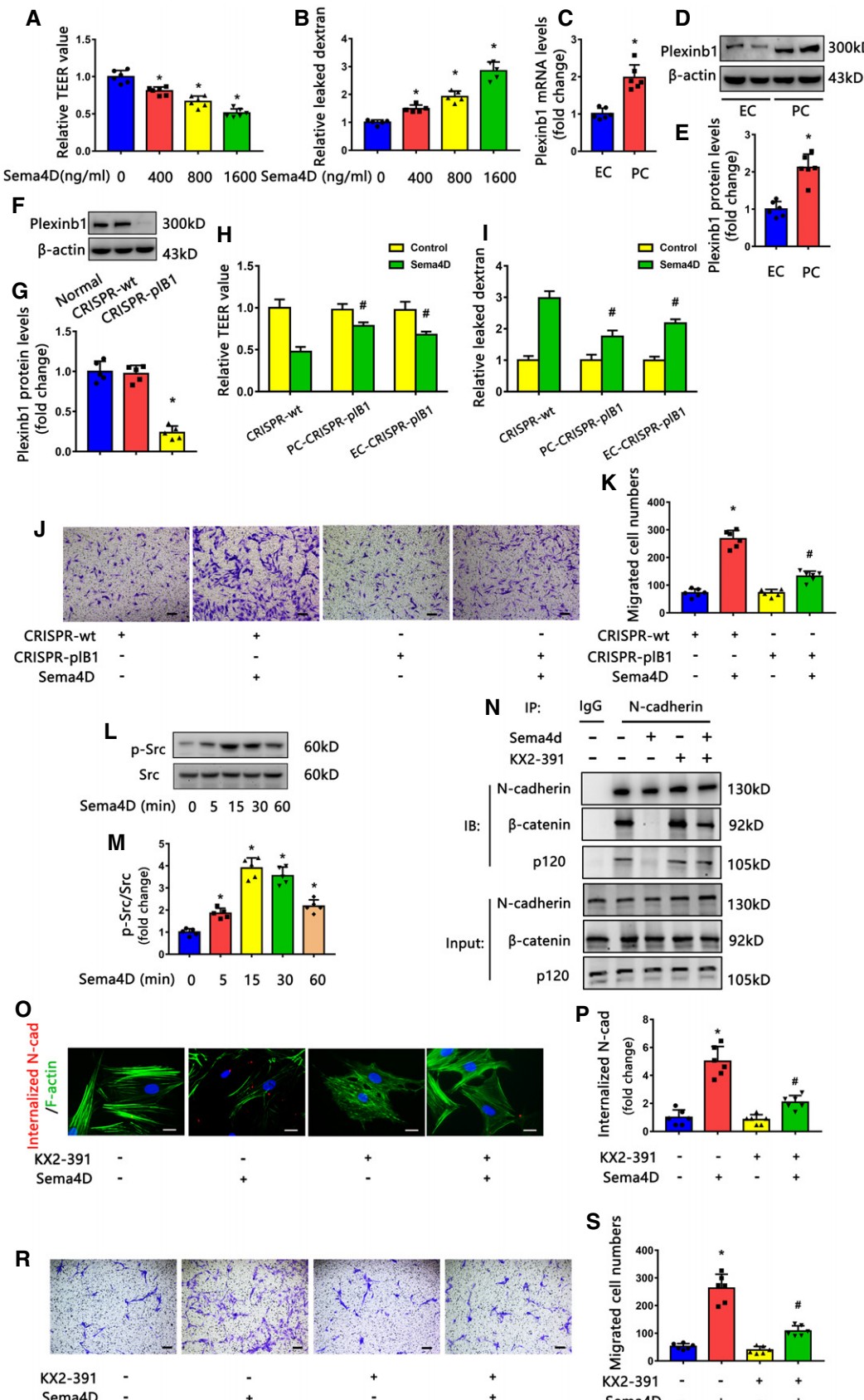

**Figure 5.**

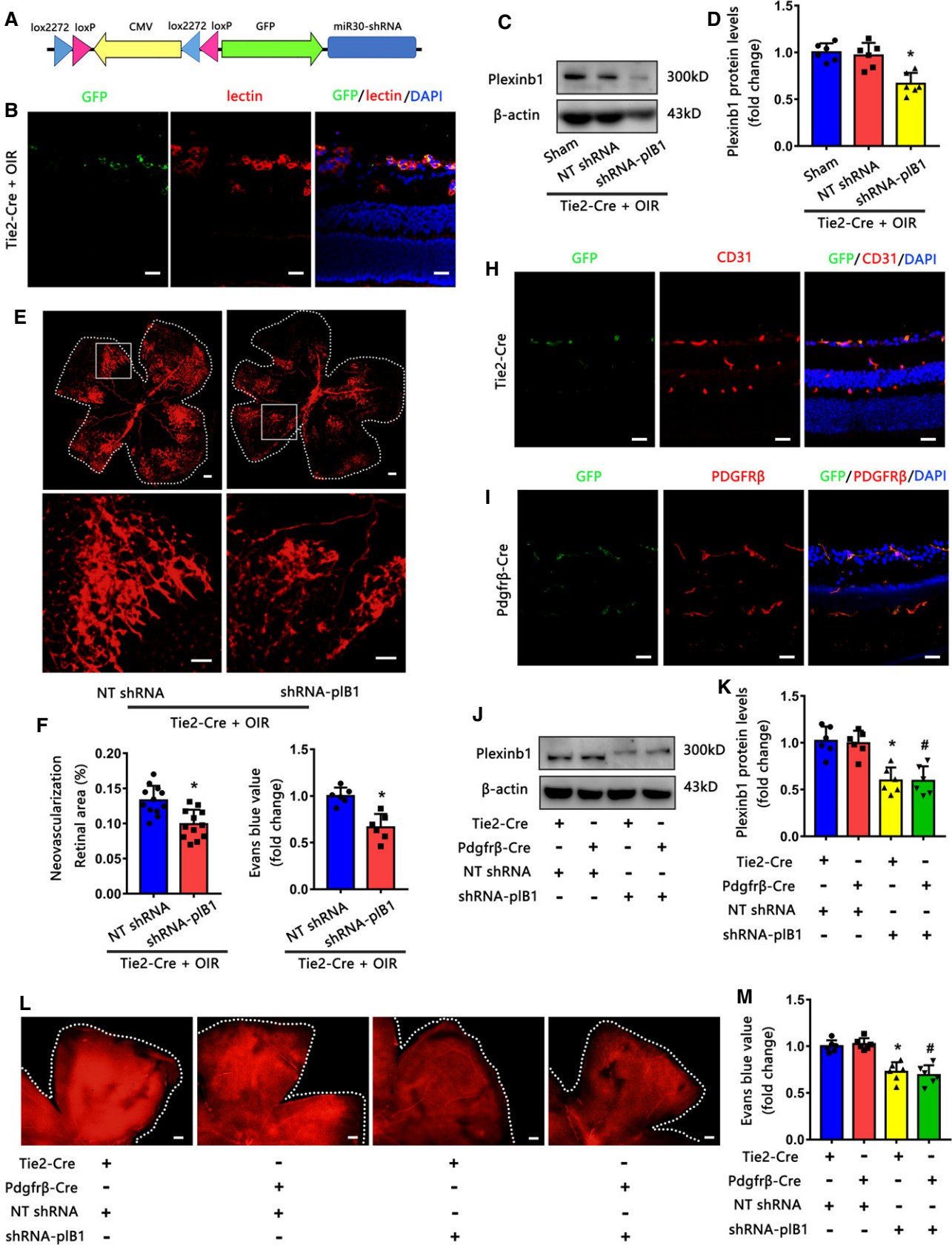

**Figure 6.**

**Figure 6. Knockdown of PlexinB1 alleviates pathologic retinal neovascularization and vascular leakage in vivo.**

A    Schematic illustration of the double-floxed Cre-inducible GFP and mir30-shRNA adenovirus (Ad).

B    GFP staining showed that the shRNA was co-localized with lectin (red) in OIR retinas (n = 5, scale bars indicate 20 μm).

C, D    Western blots (C) and quantification (D) of PlexinB1 protein levels confirmed the knockdown efficiency of Ad-shRNA-plB1 in OIR retinas (n = 6, *P < 0.05 compared with NT shRNA group).

E–G    Knockdown of PlexinB1 in endothelial cells reduced pathologic retinal neovascularization and vascular leakage in OIR model. The upper images are representative composite images of entire retinas for isolectin B4. The lower images are a cropped enlarged view from the upper entire retina images. (n = 12 in F, n = 6 in G, scale bars indicate 200 μm for upper pictures and 100 μm for lower pictures, *P < 0.05 compared with NT shRNA group).

H    GFP staining showed that the shRNA was co-localized with CD31 (red) in STZ retinas of Tie2-Cre mice (n = 5, scale bars indicate 20 μm).

I    GFP staining showed that the shRNA was co-localized with PDGFRβ (red) in STZ retinas of PDGFRβ-Cre mice (n = 5, scale bars indicate 20 μm).

J, K    Western blots (J) and quantification (K) of PlexinB1 protein levels confirmed the knockdown efficiency of Ad-shRNA-plB1 in the STZ retinas of Tie2-Cre mice and PDGFRβ-Cre mice (n = 6, *P < 0.05 compared with Tie2-Cre + NT shRNA group, #P < 0.05 compared with PDGFRβ-Cre + NT shRNA group).

L, M    Evans blue assays showed that the knockdown of PlexinB1 in endothelial cells or pericytes, respectively, alleviated vascular leakage in the STZ model. Representative Evans blue fluorescent images are shown. The calculated extracted Evans blue values were used for quantification (n = 6, scale bars indicate 200 μm, *P < 0.05 compared with Tie2-Cre + NT shRNA group, #P < 0.05 compared with PDGFRβ-Cre + NT shRNA group).

Data information: Data are means ± SD. Statistical test and P-values are reported in Appendix Table S3.
Source data are available online for this figure.

loss in DR; the suppression of Sema4D/PlexinB1 signaling prevents DR-related pathologies. To validate the therapeutic potential of anti-Sema4D/PlexinB1 signaling, we examined the effects of a Sema4D-specific neutralizing antibody (anti-Sema4D) on pathologic retinal neovascularization and vascular leakage in both the OIR and STZ DR models. Indeed, our results showed that one single intravitreal injection (IVI) of anti-Sema4D resulted in a dose-dependent inhibition of pathologic retinal neovascularization and vascular leakage in the OIR model (Fig 7A and B). The specificity of anti-Sema4D was verified by demonstrating that the pathologic retinal neovascularization and vascular leakage in the OIR model were not reduced in Sema4D-KO mice by intravitreal injection of the antibody (Fig 7C and D). We next compared the effect of anti-Sema4D with anti-VEGF neutralizing antibody (anti-VEGF), one of the standard treatments for DR. One single dose of anti-Sema4D inhibited pathologic retinal neovascularization, vascular leakage, and the phosphorylation of Src, VE-cadherin, and Fak with a comparable effect to anti-VEGF in the OIR model (Fig 7E–K). Furthermore, one single anti-Sema4D (IVI) injection in the STZ model at 5 months after diabetes onset also reduced vascular leakage as shown at 1 week postinjection (Fig 7L and M). Strikingly, a combination of anti-Sema4D and anti-VEGF injection provided a significant improvement of the effect over that of a single injection of either in both the OIR and STZ DR models, demonstrating a synergistic effect of anti-Sema4D and anti-VEGF in DR (Fig 7E–M).

Because multiple injections of anti-VEGF are needed in clinical practice, we next evaluated the therapeutic effect of multiple injections during the long period of disease induction in the STZ model. Our results revealed that multiple injections of anti-Sema4D showed an increase of tendency toward the inhibition of vascular leakage compared with anti-VEGF alone, although the effect was not statistically significant due to variability (Fig 8A and B). However, the synergistic therapeutic effect was statistically significant in combination therapy compared with monotherapy. Interestingly, we found that multiple injections of anti-Sema4D resulted in a greater effect in terms of acellular capillary formation and pericyte loss than anti-VEGF in the STZ model (Fig 8C–F). Furthermore, anti-Sema4D and anti-VEGF seem to have had a comparable effect in ameliorating VE-cadherin continuity (Fig 8G and H), while anti-Sema4D demonstrated a stronger effect than anti-VEGF therapy in ameliorating N-cadherin coverage (Fig 8I and J). In testing the effects of VEGF with Sema4D in endothelial cells and pericytes, we noted that VEGF and

**Figure 7. Anti-Sema4D and anti-VEGF have synergistic effect in inhibiting retinal neovascularization and vascular leakage.**

A, B    In the OIR model, the mice were treated with one single dose of 0.5, 1, or 2 μg anti-Sema4D (IVI) at P12 and the retinas were harvested at P17. Isolectin B4 staining and Evans blue extraction assays showed a dose-dependent anti-Sema4D effect in inhibiting pathological retinal neovascularization and vascular leakage (n = 10 for PBS, IgG, 2 μg anti-Sema4D in A. n = 12 for 0.5, 1 μg anti-Sema4D in A. n = 12 in B, *P < 0.05 compared with IgG group).

C, D    In the OIR model, the WT or Sema4D-KO mice were treated with one single dose of 2 μg IgG or anti-Sema4D (IVI) at P12 and the retinas were harvested at P17. Isolectin B4 staining and Evans blue assays were performed (n = 13, 12, 12, 11 in C; n = 12, 11, 13, 12 in D, *P < 0.05 compared with IgG + WT group, NS means no statistical significance).

E–K    In the OIR model, the mice were treated with one single dose of anti-Sema4D (2 μg), anti-VEGF (2 μg) alone or in combination (IVI) at P12 and the retinas were harvested at P17. (E and F) Isolectin B4 staining shows the neovascularization. The upper images are representative composite images of entire retinas. The lower images are a cropped enlarged view from the upper entire retina images. (n = 10, 10, 13, 10, 12 in F; scale bars indicate 200 μm for upper pictures and 100 μm for lower pictures). (G) Evans blue assays were used to quantify the vascular leakage (n = 6). (H and I) Retinal cross-sections indicated the pre-retinal neovascular cells (n = 10. Arrows indicate pre-retinal neovascular cells. Scale bars indicate 20 μm). (J and K) Western blotting detected the phosphorylation of Src, VE-cadherin, and Fak (n = 5) with the corresponding antibodies (NS means no statistical significance, *P < 0.05 compared with IgG group, #P < 0.05 compared with anti-VEGF group).

L, M    Evans blue assays showed that one single dose of anti-Sema4D (2 μg), anti-VEGF (2 μg) alone, or in combination (IVI.) attenuated pathological vascular leakage 1 week after injection in a 5-month STZ model. Representative Evans blue fluorescent images are shown. The calculated extracted Evans blue values were used for quantification (n = 6, scale bars indicate 200 μm), NS means no statistical significance, *P < 0.05 compared with IgG group, #P < 0.05 compared with anti-VEGF group.

Data information: Data are means ± SD. Statistical test and P-values are reported in Appendix Table S3.
Source data are available online for this figure.

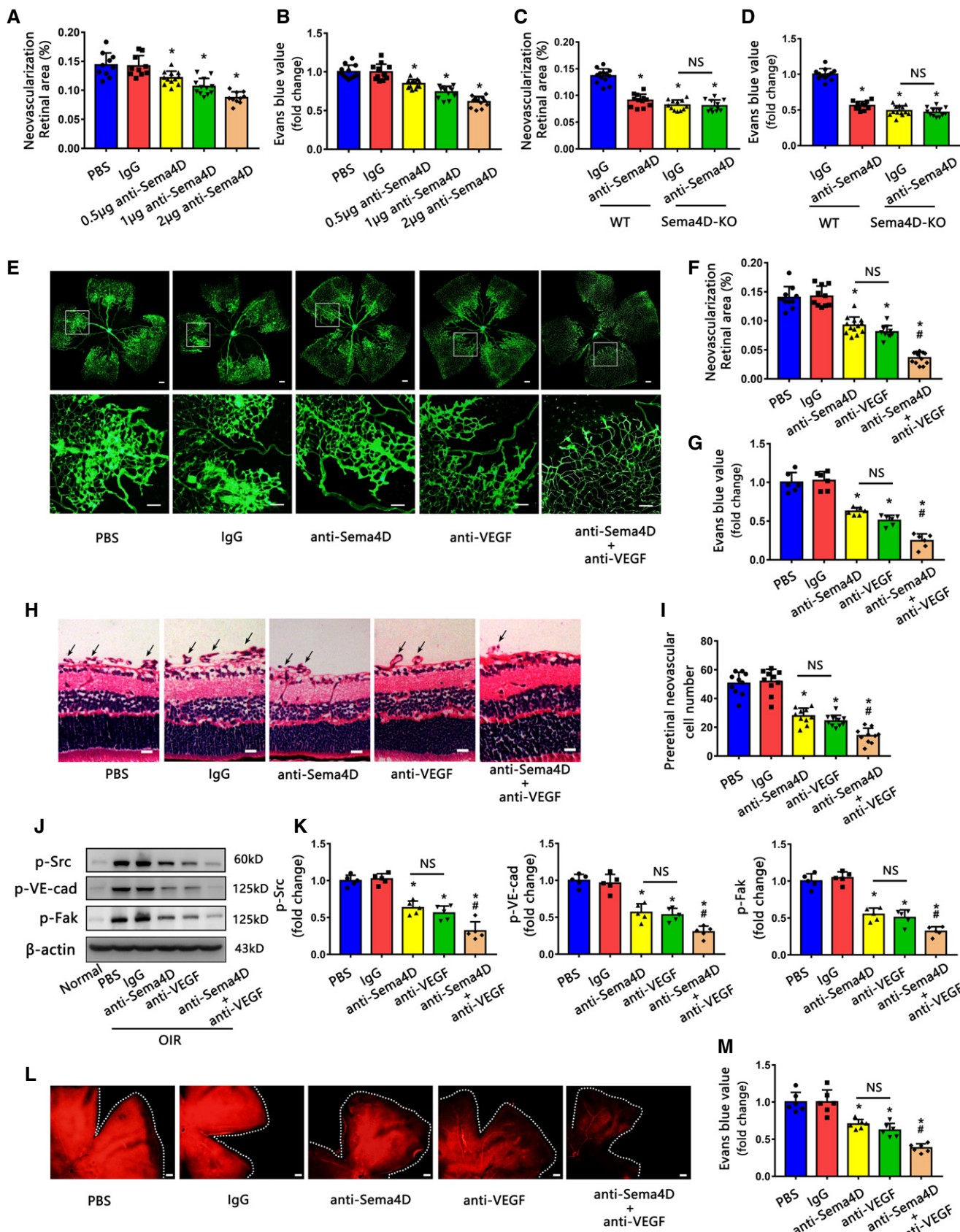

Figure 7.

Sema4D showed a comparable effect on VE-cadherin/p120-catenin/β-catenin dissociation and VE-cadherin internalization in endothelial cells (Appendix Fig S6A and B). However, Sema4D demonstrated a greater effect on N-cadherin/p120-catenin/β-catenin dissociation and N-cadherin internalization in pericytes than VEGF (Appendix Fig S6C and D). Meanwhile, we found that Sema4D displayed an additional effect than VEGF on N-cadherin expression of pericytes *in vitro* (Appendix Fig S6E–K).

In support of an additional pathogenic effect of Sema4D compared to VEGF, we found that multiple injections of anti-Sema4D showed a greater effect in alleviating vascular leakage than anti-VEGF as measured at 2 weeks after the last treatment (Fig 8K and L). Again, the combination of anti-Sema4D and anti-VEGF also showed a better therapeutic effect over monotherapy (Fig 8K and L).

## Discussion

The non- or poor response to anti-VEGF treatment among DR patients presents a therapeutic challenge. Our current study reports that an aberrant increase of Sema4D/PlexinB1 signaling not only leads to migration and permeability in endothelial cells but also promotes the N-cadherin internalization and pericyte loss that worsen the vascular permeability in DR animal models (Fig 9). Consistently, sSema4D levels are elevated in the aqueous fluid of DR patients, which may explain the poor response to anti-VEGF therapy. We demonstrate that genetic disruption of Sema4D/PlexinB1 or anti-Sema4D treatment alleviates pathologic retinal neovascularization in an OIR model as well as attenuates pericyte loss and vascular leakage in an STZ model. Furthermore, Sema4D showed an additional effect on pericyte N-cadherin dysfunction in comparison with VEGF, which underlies the therapeutic advantage of anti-Sema4D over anti-VEGF following multiple injections. Moreover, a combination of anti-Sema4D and anti-VEGF treatment provides a synergistic therapeutic effect in DR models. Therefore, our study reveals the therapeutic potential of targeting Sema4D/PlexinB1 signaling in the treatment of DR especially for those non- or poor responders to anti-VEGF.

While the intravitreal injection of anti-VEGF provides benefits to many patients suffering from DME and PDR (Nguyen *et al*, 2009;

Diabetic Retinopathy Clinical Research N *et al*, 2015; Sivaprasad *et al*, 2017), some patients are non- or poor responders to the current anti-VEGF therapy (Channa *et al*, 2014; Ashraf *et al*, 2016). We found that the levels of sSema4D in the aqueous fluid of DR patients correlated negatively with the success of anti-VEGF therapy during clinical follow-up. This indicates that sSema4D levels in aqueous fluid could help in the early identification of non- or poor responders. This is particularly important given the huge cost partly attributed to more frequent injections for non- or poor responders. Importantly, our study also indicates an alternative therapeutic strategy with anti-Sema4D to complement or improve anti-VEGF treatment in DR.

Although one single dose of anti-Sema4D showed a comparable effect with anti-VEGF in the symptomatic treatment of pathologic retinal neovascularization and vascular leakage, multiple injections of anti-Sema4D displayed better effects on alleviating acellular capillary formation and pericyte loss in comparison with anti-VEGF in the STZ model. Furthermore, a combination therapy of both provided a relief of symptoms and pathology compared to the single therapy. Moreover, our study shows that the anti-Sema4D exhibited a greater inhibitory effect than anti-VEGF in STZ-induced vascular leakage 2 weeks after the end of multiple injections. We propose that, if applied to DR, anti-Sema4D may reduce the frequency of as-needed injections thus decreasing costs in comparison with anti-VEGF monotherapy. While displaying a comparable effect with VEGF on endothelial VE-cadherin dysfunction, Sema4D showed an additional effect on pericyte N-cadherin dysfunction in comparison with VEGF. This underlies the therapeutic advantage of anti-Sema4D over anti-VEGF. Our study therefore indicates that future study should be directed toward simultaneously targeting endothelial cells and pericytes for effective DR therapy.

In addition, in contrast to VEGF, which is essential for normal retinal vascular homeostasis, Sema4D was not important for retinal vascular development, as Sema4D-KO mice displayed no obvious alterations in retinal vascular development compared to WT controls. This result suggests that anti-Sema4D also has an advantage over anti-VEGF in the form of reduced non-desirable effects. Retinal neuronal dysfunction is also one aspect of DR (Kern & Barber, 2008; Zhang *et al*, 2013; Duh *et al*, 2017; Fu *et al*, 2018). Previous studies have indicated that anti-VEGF intravitreal injection

**Figure 8. Multiple injections of anti-Sema4D alleviate acellular capillary formation and pericyte loss in the STZ model.** ▶

In the STZ model, the mice were injected weekly with anti-Sema4D, anti-VEGF alone, or a combination of the two (IVI) (A–J). Starting from 4 months after DM onset, the retinas were harvested 1 week after the fifth injection. NS means no statistical significance, *P < 0.05 compared with IgG group, #P < 0.05 compared with anti-Sema4D group.

A, B Evans blue assays were used to quantify vascular leakage in retinas. Representative Evans blue fluorescent images are shown. The calculated extracted Evans blue values were used for quantification. (n = 12, scale bars indicate 200 μm). *P < 0.05 compared with IgG group, #P < 0.05 compared with anti-Sema4D group.

C, D Acellular capillaries were counted after retinal trypsin digestion (n = 6. Arrows indicate acellular capillaries. Scale bars indicate 20 μm). *P < 0.05 compared with IgG group, #P < 0.05 compared with anti-Sema4D group.

E, F Quantification of pericyte number and PDGFRβ coverage after immunofluorescence staining of PDGFRβ and collagen IV (n = 6). *P < 0.05 compared with IgG group, #P < 0.05 compared with anti-Sema4D group.

G, H Immunofluorescence staining of VE-cadherin (green) and collagen IV (red) demonstrated VE-cadherin continuity (n = 6, scale bars indicate 10 μm). *P < 0.05 compared with IgG group, #P < 0.05 compared with anti-Sema4D group.

I, J Immunofluorescence staining of N-cadherin (green) and collagen IV (red) demonstrated N-cadherin coverage (n = 6, scale bars indicate 10 μm). *P < 0.05 compared with IgG group.

K, L The retinas were harvested 2 weeks after the fifth injection, and vascular leakage was quantified by Evans blue assays. Representative Evans blue fluorescent images are shown. The calculated extracted Evans blue values were used for quantification (n = 12, scale bars indicate 200 μm). *P < 0.05 compared with IgG group, #P < 0.05 compared with anti-Sema4D group.

Data information: Data are means ± SD. Statistical test and *P*-values are reported in Appendix Table S3.

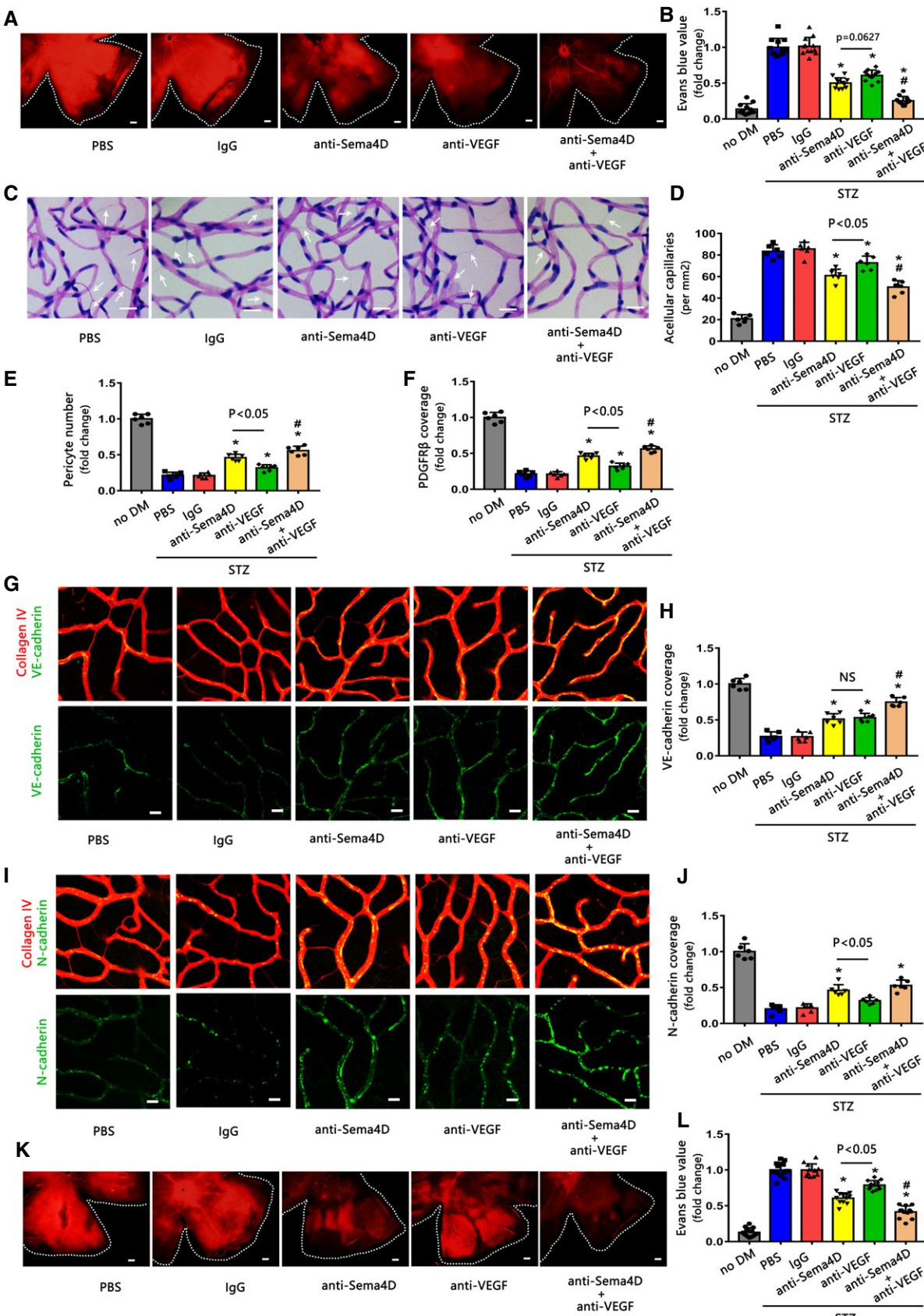

**Figure 8.**

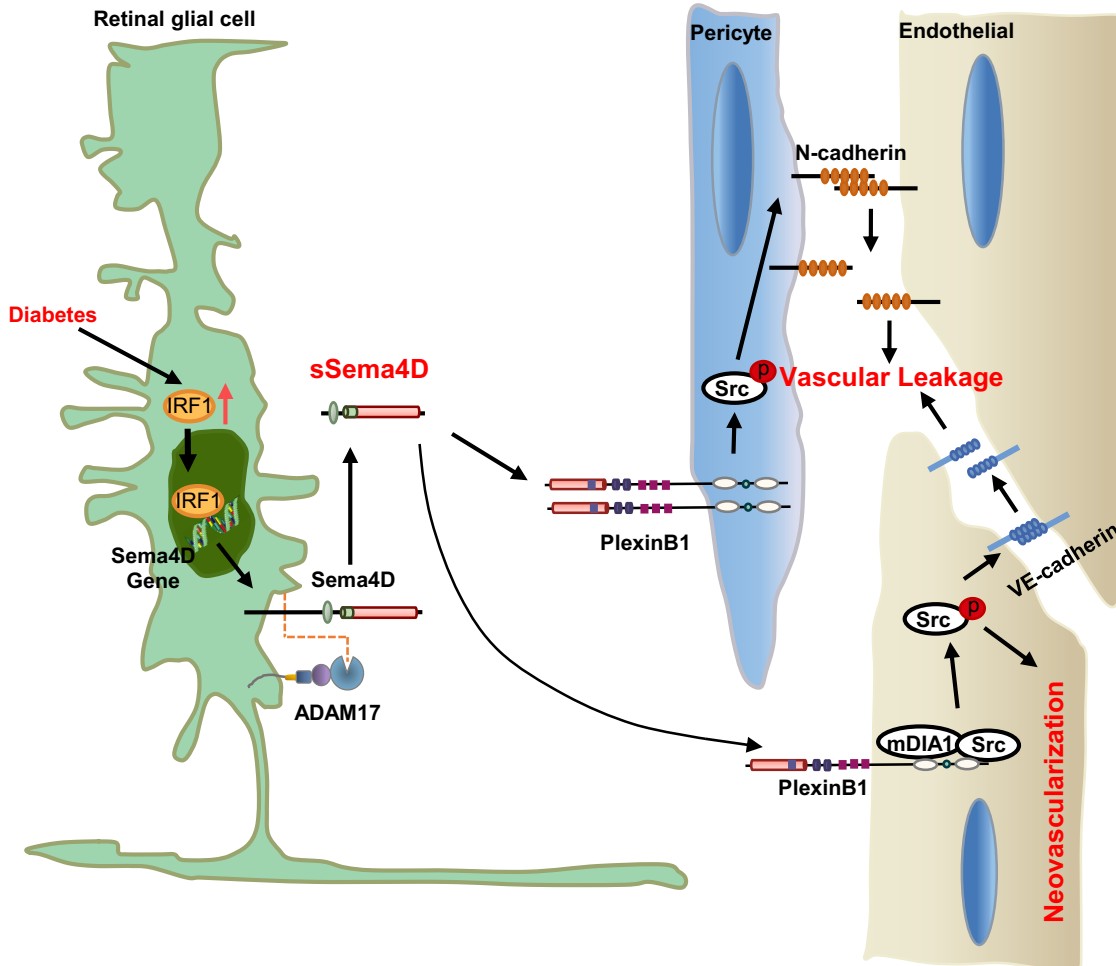

**Figure 9. A schematic illustration depicting the molecular mechanism of Sema4D/PlexinB1 signaling in DR.**

has detrimental effects on retinal neuronal function (Park *et al*, 2014a; Hombrebueno *et al*, 2015; Bucher *et al*, 2017). A recent study detected that Sema4D/PlexinB1 signaling had a novel physiological function in photoreceptor outer segment phagocytosis by retinal pigment epithelium (Bulloj *et al*, 2018). This process is essential for the survival and function of photoreceptors in the retina. Accordingly, comprehensive evaluations of Sema4D knockout and anti-Sema4D on retinal neuronal function are needed in the future.

Although increased Sema4D has been found in many diseases (Sun *et al*, 2009; Ke *et al*, 2017; Zhou *et al*, 2018b), the regulatory mechanisms remain largely unknown. Our study uncovered a novel transcription factor, IRF1, to regulate its expression in the disease condition. A previous study indicated that the gene for Sema4D also contained HIF-1α response elements, which induced its expression in tumors (Sun *et al*, 2009). Therefore, both HIF-1α and IRF1 may be involved in the regulation of Sema4D in the late stage of DR.

The adherens junctions and tight junctions of endothelial cells both play important roles in the maintenance of vascular integrity (Hartsock & Nelson, 2008; Giannotta *et al*, 2013). VE-cadherin is a component of endothelial cell-to-cell adherens junctions. In the

present study, we identified that Sema4D/PlexinB1 induced endothelial cell VE-cadherin dysfunction through the mDIA1-Src pathway. Meanwhile, Sema4D knockout and anti-Sema4D could ameliorate VE-cadherin dysfunction in the DR model. However, previous studies have reported that SEMA4D has a well-documented role in the regulation of the actin cytoskeleton (Tasaka *et al*, 2012; Bulloj *et al*, 2018). It is known that adherens junctions and tight junctions have complicated interactions with actin cytoskeleton (Hartsock & Nelson, 2008; Giannotta *et al*, 2013). It is possible that Sema4D may also induce VE-cadherin or other junction dysfunction though regulation of the actin cytoskeleton in our disease context and this needs to be further examined.

Sema4D, as a member of the semaphorin family, is a multifunctional protein that plays important roles in the immune response (Yoshida *et al*, 2015), cancer progression (Evans *et al*, 2015), neuroinflammation, and neurodegeneration (Smith *et al*, 2014; Southwell *et al*, 2015; Zhou *et al*, 2018b). The inhibition of SEMA4D/PlexinB1 signaling has been found to show clear therapeutic effects in multiple types of tumors (Evans *et al*, 2015), rheumatoid arthritis (Yoshida *et al*, 2015), multiple sclerosis (Smith *et al*, 2014), Huntington's disease (Southwell *et al*, 2015), and stroke

(Zhou *et al*, 2018b) in preclinical studies. The biotech company Vaccinex has generated a humanized IgG4 monoclonal antibody, VX15/2503, which specifically blocks the binding of SEMA4D to its receptors with a high affinity in nonhuman primates and humans (Fisher *et al*, 2016). Two phase 1 clinical trials with this antibody were successfully completed in advanced solid tumor patients (Patnaik *et al*, 2015) (Clintrials.gov identifier NCT01313065) and multiple sclerosis patients (LaGanke *et al*, 2017) (Clintrials.gov identifier NCT01764737). These clinical trials demonstrated that VX15/2503 was safe and well tolerated versus placebo at various doses when administered to patients intravenously. This antibody could be also tested for its therapeutic effect (alone or combined with anti-VEGF) in DR patients in the future, particularly in those patients who are refractory to anti-VEGF.

Our results showed that another member of semaphorin, Sema3A, was increased in the two DR models. A previous study reported that Sema3A was induced in early phases of an STZ model within the neuronal retina to trigger vascular leakage (Cerani *et al*, 2013). Sema3A could induce Src activation and VE-cadherin internalization (Le Guelte *et al*, 2012), so the contribution of Sema3A to retinal leakage should not be underestimated. However, its role in pathologic retinal neovascularization varied (Joyal *et al*, 2011; Yu *et al*, 2013; Dejda *et al*, 2014). Two studies indicated that Sema3A formed a repulsive barrier to misdirect endothelial cells (Joyal *et al*, 2011) and attracted mononuclear phagocytes (Dejda *et al*, 2014) to induce pathological pre-retinal neovascularization. Another study suggested that Sema3A could inhibit the angiogenic features of endothelial cells to reduce neovascularization in an OIR model (Yu *et al*, 2013). Therefore, semaphorins may play different roles as DR progresses. Whether synergistic regulation or the precise intervention of different semaphorins according to the disease stage provides more benefits remains to be elucidated.

Finally, because pathologic neovascularization or vascular leakage is a common mechanism in several ocular vasculopathies, such as retinopathy of prematurity, retinal vein occlusions, and neovascular AMD, anti-VEGF therapy has been explored or used as a treatment for more than one ocular vasculopathy (Campochiaro *et al*, 2016). Future research effort should be directed at determining whether Sema4D/PlexinB1 inhibition is also a useful therapy for other ocular neovascular diseases owing to its beneficial impact on pathologic neovascularization and vascular leakage.

# Materials and Methods

### Mice

All animal procedures were performed in accordance with the Guide for the Care and Use of Laboratory Animals (NIH Publication, 8th edition, 2011) and were approved by the institutional animal care and use committee, the medical ethics committee of Tongji Medical College, Huazhong University of Science and Technology, Wuhan, China.

Sema4D-knockout (Sema4D-KO) mice were generated by using CRISPR/Cas9 system as previously described (Zhao *et al*, 2016). The guide RNA (sgRNA) was designed online (http://crispr.mit.edu) around the region of previous reported Sema4D-KO mouse (Shi *et al*, 2000). In brief, a pair of sgRNA and Cas9 mRNA was microinjected into embryos to gain F0, and then, 8-week-old F0 were mated to get offspring. Genomic DNA identifying by PCR analysis was performed with primers: 5′-TCTGGGGCTCTAAGAGGT CCTT-3′; 5′- AGCCACTGAGGTCACATACACC-3′.

The Tie2-Cre mice (Cre expressed under the control of the Tie2 promoter) were obtained from the Animal Model Research Center at Nanjing University. The Pdgfrβ-Cre mice (Cre expressed under the control of the Pdgfrβ promoter) were obtained from Beijing Biocytogen. All genotypes were confirmed by PCR analysis and identified by sequencing. Animals were housed in a room with constant humidity (40–60%) and temperature (20–25°C) under a 12-h light–dark cycle. Mice were given free access to food and water.

### Oxygen-induced retinopathy model

OIR model was performed as previously described (Connor *et al*, 2009; Stahl *et al*, 2010). Briefly, mouse pups with their mother were exposed to 75% oxygen at postnatal day P7–P12 and followed by room air for a further 5 days. At P17, mice were anesthetized, and retinas were harvested. The whole-mount retinas were visualized by isolectin B4 staining. For isolectin B4 staining (Figs 3A, 6E and 7E, Appendix Fig S3D), multiple overlapping (10–20% overlap) images were obtained with a 4× lens on a fluorescence microscope. The images were imported and merged together in Adobe Photoshop to visualize the entire retinas according to previous studies (Connor *et al*, 2009; Liu *et al*, 2019). Relative pathologic neovascularization areas to total retinas areas were quantified using ImageJ software. Intravitreal injections were performed with Hamilton syringe with a 33-gauge sharp tip. For antibodies, the OIR mice were anesthetized at P12 and received intravitreal injections of anti-Sema4D neutralizing antibody (2 μg per eye, BMA-12), anti-VEGF neutralizing antibody (2 μg per eye, AF-493-NA, R&D Systems) alone or combinedly. For adenoviruses in OIR model, adenoviruses (1 μl) containing double-floxed Cre-inducible GFP and mir30-shRNA were intravitreally injected into Tie2-Cre mice at P7 before exposure to 75% O2, the retinas were harvested at P17. The animals were randomized into different treatments. The wild-type or knockout mice were randomly allocated to experimental groups. For intravitreal injections, the mice were treated with antibiotics to prevent infection. Those with eyeball infections after injection were excluded.

For pre-retinal neovascular cell quantification, eyeballs were fixed in 4% paraformaldehyde (PFA) for 24 h and embedded in paraffin for retinal cross-sections (10 μm). Sections crossing the optic nerve were deparaffinized and stained with hematoxylin and eosin (HE). The pre-retinal neovascular cells were quantified from each group in a masked fashion.

### Streptozotocin-induced diabetic mice

Eight-week-old male C57BL/6 mice were starved for 4 h before streptozotocin (STZ, Sigma, St. Louis, MO) injection. They then received an intraperitoneal injection of STZ (50 mg/kg) or citrate buffer (vehicle control) for 5 days consecutively. The fasting blood glucose of the mice was evaluated at 7 days after the last STZ injection by a glucometer (ONETOUCH, Johnson & Johnson, USA). Mice with glucose levels of 16.7 mM or above were deemed diabetic. The fasting blood glucose was continuously monitored per month to ensure the models were successful (those failed mice were

excluded). For antibodies, mice were received intravitreal injections of anti-Sema4D neutralizing antibody (2 μg per eye, BMA-12), anti-VEGF neutralizing antibody (2 μg per eye, R&D Systems) alone or combinedly in the indicated time. For adenoviruses in STZ model, adenoviruses (1 μl) containing double-floxed Cre-inducible GFP and mir30-shRNA were intravitreally injected into Tie2-Cre and Pdgfrβ-Cre mice at the beginning of diabetes induction and were repeated once a month, and the retinas were harvested at 2 months of diabetes. The animals were randomized into different treatments. The wild-type or knockout mice were randomly allocated to experimental groups. For intravitreal injections, the mice were treated with antibiotics to prevent infection. Those with eyeball infections after injection were excluded.

## Evans blue assays

For the STZ mice, Evans blue dye (45 mg/kg) was injected through the tail vein and circulated for 2 h. For the OIR mice, Evans blue dye (200 mg/kg) was intraperitoneally injected and circulated for 4 h at P17. The mice were intracardially perfused, both eyes were fixed with 4% PFA for 2 h, and the retinas were dissociated. Some retinas were used for Evans blue fluorescent detection with a fluorescence microscope. Other retinas were incubated with formamide overnight at 70°C to extract Evans blue dye. Absorbance of the extracts at 620 and 740 nm (background) were detected with a microplate reader. The concentrations were calculated from a standard curve and were normalized to tissue weight and Evans blue blood levels. The calculated extracted Evans blue values were used for quantification. One data point represented the mean value of three technical replicates from one independent biological replicate.

## Viral vectors

The lentivirus containing CRISPR/Cas9-mediated depletion of plexinB1 (CRISPR-plB1) or negative control (CRISPR-wt) was commercially purchased from GeneChem company (Shanghai China). Briefly, the online CRISPR design tool (http://crispr.mit.edu) was used to predict the sgRNA sequences for the mouse PlexinB1 gene. The targeting sequence for plexinB1 used in this study was 5′-GTCGCCGGTCGGAGTGCTCA -3′. DNA oligos were synthesized and cloned into vectors containing CAS9. Then the vector was packaged into lentivirus. After purification of the CRISPR/Cas9 lentivirus, the lentivirus was transfected into the targeted cells according to the manufacturer's instructions. The efficiency was determined by Western blot analysis. Because the primary mouse microvascular brain endothelial cells and pericytes have a limited passage number and are difficult to be clonally expanded, a pool of knockdown cells were used in our study.

To specific knockdown of plexinB1 in endothelial cells and pericytes *in vivo*, an adenovirus containing double-floxed Cre-inducible GFP and mir30-shRNA targeted against PlexinB1 was constructed. The sequence of PlexinB1 shRNA was 5′-GCACACATCTACAA CACTT-3′. The adenovirus was intravitreally injected into Tie2-Cre and PDGFRβ-Cre mice to specifically express shRNA in endothelial cells and pericytes *in vivo* respectively. The specificity was confirmed by GFP staining with endothelial cell and pericyte markers. The efficiency was determined by Western blot analysis.

## Western blot analysis

For cultured cells, cells were washed with ice-cold PBS and lysed with a RIPA lysis buffer containing freshly added protease cocktail inhibitors for 30 min at 4°C. For phosphorylation detection, the cells were starved in DMEM with 0.5% FBS before different stimulations. For retinas, mice were intracardially perfused, and the retinas were freshly dissociated in ice-cold PBS. Four same-treated retinas were mixed as one independent biological replicate. Retinas were cut into pieces and lysed with a RIPA lysis buffer containing freshly added protease cocktail inhibitors for 30 min at 4°C. The lysate was sonicated and centrifuged at 12,000 g for 15 min at 4°C, and then, the supernatant was collected. Protein concentrations were quantified by BCA Protein Assay Kit (Beyotime Biotechnology, Shanghai, China). Equal amount of proteins was separated by SDS–PAGE, transferred onto PVDF membranes and detect with the following primary antibodies: SEMA4D antibody (1:500, AF5235, R&D Systems), PlexinB1 antibody (1:500, ab90087, Abcam), PlexinB1 antibody (1:500, ab39717, Abcam), IRF1 antibody (1:500, #8478, Cell Signaling), and ADAM17 antibody (1:1,000, ab2051, Abcam), ADAM10 antibody (1:1,000, ab124695, Abcam), MMP14 antibody (1:1,000, ab53712, Abcam), ADAMTS4 antibody (1:1,000, ab185722, Abcam), mDIA1 (1:1,000, ab129167, Abcam), p-FAK (1:500, Tyr576/577, #3281, Cell Signaling), FAK (1:1,000, #3285, Cell Signaling), p-Src antibody (1:500, Tyr416, #6943, Cell Signaling), Src antibody (1:1,000, #2108, Cell Signaling), Src antibody (1:1,000, 60315-1-Ig, Proteintech), p-VE-cadherin (1:500, Y685, ab119785, Abcam), VE-cadherin (1:500, sc-9989, Santa Cruz), VE-cadherin (1:500, AF1002, R&D Systems), N-cadherin (1:1,000, 33-3900, Life Technologies), N-cadherin (1:500, ab18203, Abcam), N-cadherin (1:500, AF6426, R&D Systems), β-catenin (1:500, #8480, Cell Signaling), β-catenin (1:1,000, 51067-2-AP, Proteintech), p120-catenin (1:1,000, 12180-1-AP, Proteintech), and β-actin (1:2,000, A01010, Abbkine). The membranes were incubated with horseradish peroxidase-conjugated secondary antibodies and were visualized by ECL solution with a BioSpectrum Imaging System (UVP, USA). The intensities of the bands were analyzed by ImageJ software package (National Institutes of Health, Bethesda, MD, USA).

## Patient samples

Aqueous fluid samples of diabetic retinopathy were obtained from patients immediately before injection of intravitreal anti-VEGF neutralizing antibody. Control aqueous fluid samples were collected from nondiabetic patients before undergoing cataract surgery or vitrectomy surgery for epi-retinal membrane. All procedures were performed in participating patients after informed consent. These procedures were approved by an ethics committee of Tongji Medical College and strictly followed the tenets of the WMA Declaration of Helsinki and the Department of Health and Human Services Belmont Report.

## Immunofluorescence

Eyeballs were fixed with 4% PFA and embedded in paraffin or in O.C.T. Compound for retinal cross-sections (10 μm). Paraffin sections were first dewaxed. After heat-mediated antigen retrieval

with citrate buffer, the sections were permeabilized with 0.5% Triton X-100 for 10 min and blocked with 10% donkey serum for 30 min at room temperature. Then, the sections were incubated with primary antibodies at 4°C overnight. For retinal whole mount, eyeballs were fixed in 4% PFA on ice or in methanol at −20°C for 2 h. Retinas were dissected after fixation, then blocked, and permeabilized in blocking buffer (1% BSA plus 0.5% Triton X-100 in PBS) overnight at 4°C, and retinas were incubated with primary antibodies at 4°C for 5 days or overnight for isolectin B4. Primary antibodies are as follows: SEMA4D antibody (1:50, sc-390675, Santa Cruz), VE-cadherin (1:50, #555289, BD Biosciences), N-cadherin (1:50, 33-3900, Life Technologies), GFAP (1:100, ab4674, Abcam), NeuN (1:100, A11449, Abclonal), Collagen IV (1:100, #1340, Southern Biotech), Desmin (1:100, ab15200, Abcam), PDGFRβ (1:100, ab32570, Abcam), Isolectin B4 (1:100, #1201 or 1207, Vector Laboratories), CD31 (1:50, AF3628, R&D Systems), and PDGFRβ (1:50, AF1042, R&D Systems). For secondary detection, fluorescein-conjugated secondary antibodies were applied for 2 h at room temperature, and then, fluorescent images were captured using a confocal microscope.

The pericyte number was quantified from ten random fields per retina and was normalized to mm$^2$ of capillary area. The pericyte coverage, VE-cadherin continuity, and N-cadherin coverage were quantified by ImageJ software and normalized to collagen IV-positive area, and the value per retina represented the mean values from ten random fields. For two same-treated retinas of one mouse, the average value of two retinas was indicated as one data point.

## Wound closure assays

Confluent endothelial cells were starved in DMEM with 0.5% FBS overnight, and linear scratch wounds were gently generated using a sterilized 200-μl tip. After washing away the floating cells, cells were treated with different stimulations in DMEM with 0.5% FBS containing mitomycin-C for an additional 24 h. The scratch wounds were photographed at indicated time points using a microscope with a digital camera. The area of the wounds was analyzed using ImageJ.

## Trans-endothelial electrical resistance and dextran permeability

The two models used in our experiments were endothelial monolayers and Transwell-based co-culture of endothelial cells with pericytes. For the co-culture system, transwell chambers (6.5 mm diameter polycarbonate membranes containing 0.4-μm pore, Corning, NY, USA) were coated with fibronectin for 1 h at room temperature. Endothelial cells were seeded on the lower chambers to adhere firmly overnight. Pericytes were then seeded on the upper chambers and co-cultured for a further 3 days. TEER value was measured using an Epithelial Volt/ohm Meter and STX2 electrodes (World Precision Instruments, Sarasota, FL, USA).

The permeability was also evaluated with RITC-conjugated dextran (Mw 10000; Sigma-Aldrich) as previously described. Briefly, 300 μg/ml of RITC-dextran was added to the upper chambers, 30-μl samples were obtained from the lower chambers 0, 30, 60, and 120 min after RITC-dextran was added and replaced with an equal volume of medium. The fluorescence intensity was assessed by a fluorescence microplate reader (PerkinElmer, Waltham, MA, USA).

## Retinal trypsin digestion

Retinal vasculature preparations and quantification of acellular capillaries were performed as previously described (Lai *et al*, 2017). Freshly enucleated eyes were fixed in 4% PFA for 24 h. Then, the retinas were dissociated and incubated with 3% trypsin (dissolved in pH 7.8 Tris–Hcl) at 37°C, trypsin digestion was completed when the retinas begun disintegration, and, then, the retinas received gently shaken till to free vessel network. The free vessels were mounted on adhesive glass slides for dry. The vessels were stained with periodic acid-Schiff (PAS) and hematoxylin. The acellular capillaries were quantified from ten random fields per retina, and the values were normalized to mm$^2$ of capillary area. The value represented the mean values from ten random fields.

## Cell culture and reagents

Primary mouse brain microvascular endothelial cells and pericytes were isolated and cultured as previously described with some modifications (Wan *et al*, 2018). Endothelial cells were cultured in medium 131 (Invitrogen, Carlsbad, CA, USA), and pericytes were cultured in pericyte medium (ScienCell, USA). The purity of endothelial cells was confirmed by flow analysis with anti-CD31 antibody. The purity of pericytes was confirmed by flow analysis with anti-CD13 antibody (Crouch & Doetsch, 2018). The purity of these two cells was more than 95%. The plates were pre-coated with gelatin. The bEnd.3 and HEK 293T cells were purchased from ATCC and grown in DMEM supplemented with 10% fetal bovine serum. Isolation of primary retinal glial cells was performed as previously described (Zhou *et al*, 2014a). All cells were cultured at 37°C in a humidified 5% CO$_2$ incubator. All cell lines were regularly authenticated by their marker genes. Mycoplasma contamination was routinely checked using Lookout Mycoplasma PCR Detection Kit (Sigma, St. Louis, MO). KX2-391, GSK2256098, and TAPI-1 were purchased from Selleck Chemicals.

## siRNA transfection

The non-target siRNA (NT siRNA) and IRF1, mDIA1, ADAM17, ADAM10, MMP14, and ADAMTS4 siRNA were commercially obtained from RiboBio company (Guangzhou, China). The sequences were included in Appendix Table S1. The siRNAs were transfected into cells following the manufacturer's protocol of Lipofectamine 2000 (Invitrogen). The silencing efficiency was detected by Western blotting at 24 h after transfection.

## RNA isolation and quantitative real-time PCR

Total RNAs were extracted from cell lysates and retinas with RNA isoplus kit (Takara, Kyoto, Japan). Two same-treated retinas were mixed as one independent biological replicate for RNA extraction. cDNAs were synthesized with the cDNA Synthesis Kit (Takara). Real-time PCR was performed using SYBR Premix Ex TaqTM Kit (Takara) following the manuals. Primers for PCR were in Appendix Table S2.

## Immunoprecipitation

Cells were lysed in immunoprecipitation buffer (Beyotime Biotechnology, Shanghai, China), and equal amounts of 500 μg protein were incubated with specific antibodies against PlexinB1 (ab90087, Abcam), N-cadherin (ab18203, Abcam), and VE-cadherin (ab33168, Abcam) overnight at 4°C with gentle rotation. 40 μl Protein A&G agarose beads (Beyotime Biotechnology) were then added and incubated for another 2 h. Agarose beads were precipitated to bottom by transient centrifugation at 4°C and then washed three times with the lysis buffer. The complexes were eluted from the beads by heating samples in loading buffer with SDS and then used for Western blotting.

## ELISA

Aqueous fluid samples were centrifuged at 4,000 g for 5 min at 4°C to remove fragments, and then, supernatants were collected, and sterile protease cocktail inhibitors were added to prevent protein degradation. The aqueous fluid samples were stored at −80°C. Cell culture conditional medium was collected from primary glial cells after indicated stimulations and centrifuged to remove cell fragments.

Soluble Sema4D (sSema4D) levels in aqueous fluid samples and cell culture conditional medium were measured using ELISA kits (MyBioSource) according to manufacturer's instructions.

## Chromatin immunoprecipitation (ChIP) assays

ChIP assays were performed as previously described (Wu *et al*, 2017). Glial cells were fixed with 1% formaldehyde for 10 min and quenched with 125 mM glycine. The cells were washed with ice-cold PBS and lysed with lysis buffer containing various protease inhibitors. The chromatin in the lysates was sonicated into 100–500 bp DNA fragments and was centrifuged to remove the residue. The supernatants were transferred to a new tube and immunoprecipitated by an IgG antibody (negative control) or an IRF1 antibody (Cell Signaling). The immunoprecipitated DNA was extracted with a DNA purification kit and amplified by PCR using the following primer pairs: 5′-GGTCAAGGTCTCTCACCTGTT-3′; 5′-GGGCTTGCTT CCAAGTTGAC-3′.

## Fluorescence *in situ* hybridization

Sema4D mRNA expression in retinal sections was detected using a fluorescent *in situ* hybridization kit (RiboBio, Guangzhou, China) according to the manufacturer's instructions. Briefly, mice were intracardially perfused, and eyes were fixed with 4% PFA (containing 0.1% diethyl pyrocarbonate) at 4°C overnight. Retinal sections were serially sectioned after dehydration. Antigen retrieval was performed with sodium citrate buffer at 90°C for 30 min followed by incubating with proteinase K for 20 min. The sections were permeabilized with PBS containing 0.5% Triton X-100 and treated with blocking/pre-hybridization buffer at 37°C for 30 min. Cy3-labeled RNA probes targeting Sema4D were resuspended in hybridization buffer and incubated on the sections at 37°C overnight. Then, the sections were washed with 4× SSCT, 2× SSC, and 1× SSC in order at 42°C. After *in situ* hybridization, the

**The paper explained**

**Problem**
Diabetic retinopathy (DR) is a common complication of diabetes that leads to blindness. Anti-vascular endothelial growth factor A (anti-VEGF) has been used as one of the standard treatments for DR. However, its therapeutic effect is limited in non- or poor responders, who experience persistent macular edema and a loss of visual acuity over time despite frequent anti-VEGF injections. Thus, it is imperative to develop improved therapies for DR patients to treat non- or poor responders.

**Results**
By performing a comprehensive analysis of the semaphorins family, we identified an increased expression of Sema4D during oxygen-induced retinopathy (OIR) and streptozotocin (STZ)-induced retinopathy. The levels of soluble Sema4D (sSema4D) were significantly increased in the aqueous fluid of DR patients and correlated negatively with the success of anti-VEGF therapy during clinical follow-up. We found that Sema4D/PlexinB1 induced endothelial cell dysfunction via mDIA1, which was mediated through Src-dependent VE-cadherin dysfunction. Furthermore, genetic disruption of Sema4D/PlexinB1 or intravitreal injection of anti-Sema4D antibody reduced pericyte loss and vascular leakage in the STZ model as well as alleviated neovascularization in the OIR model. Moreover, anti-Sema4D had a therapeutic advantage over anti-VEGF in relation to pericyte dysfunction. Anti-Sema4D and anti-VEGF also conferred a synergistic therapeutic effect in two DR models.

**Impact**
This study suggests an alternative therapeutic strategy with the inhibition of Sema4D/PlexinB1 to complement or improve the current treatment of DR.

sections were further stained with GFAP (1:100, ab4674, Abcam) followed by incubating with fluorescein-conjugated secondary antibody. Fluorescent images were captured using a confocal microscope.

## Fluorescence-activated cell sorting (FACS)

Adult mice were intracardially perfused with ice-cold PBS, and retinas were quickly dissociated from eyeballs in ice-cold PBS. The retinas were digested in 0.5 mg/ml collagenase D (Roche) and 750 U/ml DNase (Sigma) in Hanks' balanced salt solution at 37°C for 25 min with gentle shaking. The disaggregated cells were washed with ice-cold PBS (containing 3% FBS) and filtered with 70-μm mesh. Cell suspensions were blocked with anti-mouse CD16/32 (Biolegend) for 10 min at room temperature to block Fc receptors. Cells suspensions were then labeled for 30 min at 4°C with the following antibodies: APC rat Anti-mouse CD31 (BD Biosciences, 551262) and FITC rat anti-mouse CD13 (BD Biosciences, 558744). PE/Cy7 rat anti-mouse CD45 (Biolegend, 103114) was added to exclude hematopoietic lineage cells, and 7-AAD was used to label dead cells. $CD31^+/CD13^-/CD45^-/7\text{-}AAD^-$ cells were sorted as endothelial cells; $CD13^+/CD31^-/CD45^-/7\text{-}AAD^-$ cells were sorted as pericytes with Aria II sorter (BD Biosciences). The data were analyzed using FlowJo software. The sorted cells were lysed with RNA isoplus kit (Takara, Kyoto, Japan) for RNA extraction.

## Apoptosis assays

Pericytes were sub-cultured into 6-well plates overnight and stimulated with different concentrations of recombinant Sema4D for another 24 h. Then, the percentage of apoptosis cells was evaluated with an Annexin V-FITC Apoptosis Detection Kit (BD Bioscience) by flow cytometry according to the manufacturer's protocol. The data were analyzed using FlowJo software.

## Transwell assays

A 24-transwell dish (6.5 mm diameter polycarbonate membranes containing 8.0-µm pore, Corning, NY, USA) was used. Pericytes were serum-starved overnight in DMEM containing 0.5% FBS, then trypsinized, and suspended in DMEM. Pericytes ($3.0 \times 10^4$ cells/well) were added to the upper chambers with 200 µl DMEM and incubated for 6 h with 500 µl serum-free DMEM plus recombinant Sema4D in the lower chambers. The chambers were fixed with 4% PFA, and the cells were stained by crystal violet for 15 min. Then, the cells remained in the upper chambers were gently erased by a cotton bud. The cells that migrated to the bottom of the chambers were randomly counted using a light microscope with a digital camera.

## VE- and N-cadherin internalization assays

To assess VE-cadherin or N-cadherin internalization, indicated cells were incubated with antibodies against the extracellular domain of VE-cadherin (1:50, #555289, BD Biosciences) or N-cadherin (1:50, AF6426, R&D Systems) at 37°C in DMEM containing 1% BSA for 1 h. The unbound antibodies were removed with ice-cold DMEM medium. The cells were treated with recombinant Sema4D, VEGF with or without Src inhibitor at 37°C for 4 h, and fixed with 4% PFA for 20 min. An excess of anti-rat or anti-sheep IgG antibody was used to block the antibodies remaining on cell surfaces. Internalization of VE-cadherin or N-cadherin was visualized with Alexa 594-conjugated anti-rat or anti-sheep IgG antibody after the samples were permeabilized with 0.5% Triton X-100. Samples were imaged with microscope, and internalized particles were quantified from ten random fields using software.

## Statistical analysis

All data were expressed as mean ± SD. All data sets were tested for normality of distribution using the Shapiro–Wilk test. To evaluate the significant differences of the data with normal distribution, statistical analysis between two groups was performed using unpaired two-tailed Student's $t$-test. One-way ANOVA was used to statistical analysis among more than two groups, followed by a Tukey's post hoc test for multiple-group comparisons. Two-way ANOVA followed by the Tukey's test was used in Fig 5H and I, Appendix Figs S2B and S6B and D. For the data with non-normal distribution, nonparametric statistical analysis between two groups was performed by the Mann–Whitney test, and Kruskal–Wallis test was used for more than two groups followed by a Dunn's test for multiple comparisons. The correlations between sSema4D levels and changes to anti-VEGF therapy were determined by Spearman's rank test. All $n$ values represented the number of independent biological replicates in our study. Sample sizes *in vitro* and *in vivo* were chosen based on similar

studies from the literature. The experimenters were blinded to animal genotype and grouping information.

**Expanded View** for this article is available online.

## Acknowledgements

We thank other colleagues in Dr Hu's laboratory for various technological assistance and helpful suggestions during this investigation. This work was supported by National Key R&D Program of China (No. 2018YFC1312200), National Natural Science Foundation of China (No. 81601027 to Y.N.L, No. 81820108010 to B.H, No. 81571119 to B.H, No. 81571139 to L.M, No. 81771249 to Y.P.X, No. 81400969 to Q.W.H), Major refractory diseases pilot project of clinical collaboration with Chinese & Western Medicine (SATCM-20180339), and New Century Excellent Talents in University (No. NCET-10-0406 to B.H).

## Author contributions

J-hW and Y-nL performed major experiments and wrote this manuscript. BH designed the research and edited the manuscript. A-qC performed the immunofluorescence staining. C-dH performed Western blotting. C-lZ and H-lW bred the mice. P-CL and YW provided clinical samples. Y-fZ, LM, Y-pX, Q-wH, and H-jJ provided some suggestions. Z-yY edited the manuscript.

## Conflict of interest

The authors declare that they have no conflict of interest.

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
