## [Review Process File · EMBO Molecular Medicine]

Inhibition of Sema4D/PlexinB1 signaling alleviates vascular dysfunction in diabetic retinopathy

Jie-hong Wu; Ya-nan Li; An-qi Chen; Can-dong Hong; Chun-lin Zhang; Hai-ling Wang; Yi-fan Zhou; Peng-Cheng Li; Yong Wang; Ling Mao; Yuan-peng Xia; Quan-wei He; Hui-juan Jin; Zhen-yu Yue; Bo Hu

Review timeline:

Submission date:	3 December 2018
Editorial Decision:	24 January 2019
Additional correspondence:	31 January 2019
Revision received:	23 May 2019 2019
Manuscript withdrawn:	6 June 2019
Manuscript resubmitted:	2 October 2019
Editorial Decision:	13 November 2019
Revision received:	2 December 2019
Accepted:	6 December 2019

Editor: Lise Roth

Transaction Report:

1st Editorial Decision

24 January 2019

Thank you for the submission of your rebuttal letter to EMBO Molecular Medicine following our pre-consultation exercise. We have now received the referees' comments on this letter. They agree that you are in a position to address their concerns by adding experimental data and providing more information on the methods and analysis, and they would in principle support publication of your work pending major and complete revisions of the current manuscript.

Please note that referee 3 insists on using cell-specific conditional knockout mice, which was not addressed in your current rebuttal letter. Moreover, referee 1 would like you to cite and comment on the following publication: Cerini et al, Cell metabolism, 2013.

Addressing the reviewers concerns in full will be necessary for further considering the manuscript in our journal, and acceptance of the manuscript will entail a second round of review. Given the extent of the revisions, we would be ready to extend the revisions time to 6 months. EMBO Molecular Medicine encourages a single round of revision only and therefore, acceptance or rejection of the manuscript will depend on the completeness of your responses included in the next, final version of the manuscript. For this reason, and to save you from any frustrations in the end, I would strongly advise against returning an incomplete revision.

EMBO Molecular Medicine has a "scooping protection" policy, whereby similar findings that are published by others during review or revision are not a criterion for rejection. Should you decide to submit a revised version, I do ask that you get in touch after six months if you have not completed it, to update us on the status.

Please also contact us as soon as possible if similar work is published elsewhere. If other work is published, we may not be able to extend the revision period beyond six months.

I look forward to receiving your revised manuscript.

***** Reviewer's comments *****

Referee #1 (Remarks for Author):

In this study, Bo Hu and colleagues follow up on their previous data on SEMA4D in regulation of the BBB and that inhibition of SEMA4D improves the outcome of stroke in the rat (Faseb J 2018). They use a wide range of methods and models to address the role of SEMA4D in retinopathies. While some of the presented data (cleavage of SEMA4D, the role of PlexinB1 as a receptor for SEMA4D) have been presented before, there are novel data in this presentation. Of particular interest is the increased production of soluble SEMA4D in the aqueous fluid of patients with diabetic retinopathy. However, there are a number of serious concerns with this study, as listed below.

1. The variability of the data is unexpectedly small. Throughout, error bars are miniscule which calls into question whether all data points have been included and how data was processed. See for example i) the lentivirus-mediated knockdown of PlexinB1 in the retina followed by STZ-treatment for 3 months and blotting in Supplemental Figure 9 A and G. How can lentivirus-mediated delivery of CRISPR-Cas9 to the vitreous of individual mice, followed by STZ-treatment, again of individual mice, give rise to such all or none effect with very little variation?! Moreover, ii) how come that endothelial cells do not seem to grow to fill the wound at all in the control, in 24 h (see for example Fig. 4). In 24h, such dense cultures would be expected to completely fill the wound. iii) How come pSrc/pFAK levels are barely detected in tissue culture cells before they are treated with e.g. Sema4D (see Fig. 5B). Adherent cells in tissue culture have high levels of pSrc/pFAK through integrin-dependent activation. How do these cells remain attached at all? iv) The retinopathy field agrees it is challenging to detect vessel leakage from the retinal superficial vasculature. Still the authors manage to obtain 10-fold induction of leakage (see Fig. 3F-G). For the *in vivo* analyses, the authors state throughout that the n=6; I assume this means 6 mice or is it 3 mice, using each of the two eyes/mouse as individual samples (which should not be done)? The authors need to declare whether these were selected from a much larger cohort. How many times did they do individual repeats (should be 3 times). How do they explain to have a variability of less than 10% in these complicated *in vivo* analyses? In their present form, the results are not credible.

2. The authors suggest that anti-SEMA4D should be used together with anti-VEGF antibodies to treat DR, based on that SEMA4D KO mice show more limited retinal damages and that treatment of STZ-induced DR with anti-SEMA4D antibodies is beneficial. In Fig. 1 D, they show that there are elevated levels of SEMA4D in aqueous fluid of controls and DR patients, treated with anti-VEGF antibodies. Fig. 1E and F show that there is less soluble SEMA4D in patients with a more severe disease (with higher CST and MV change). The rationale for treating patients with a more severe disease with anti-SEMA4D antibodies therefore does not seem to exist, as these patients based on the cohort shown in Fig. 1E and F, have very low levels of SEMA4D. Please explain the rationale better.

3. For knocking down PlexinB1, the authors perform intravitreal injection of lentivirus (1 μ l) expressing CRISPR-wt or CRISPR-pIB1 at P7. Please give exact information on the degree of PlexinB1 knockdown in retinal endothelial cells by PCR and by immunostaining. PlexinB1 is broadly expressed in general. Show by immunostaining which cells in the retina express PlexinB1 before and after CRISPR-pIB1. Discuss how the biology still can be attributed to knockdown of PlexinB1 in endothelial cells.

4. The study comprises a lot of data; in addition to the main figure, 11 supplemental figures with many panels each, are provided. The impression is of brute force rather than assembling the presentation around a theme. The methods are very sparsely described. Some methods/results do not need to be included such as the data on Sema4D cleavage or regulation by IRF1. The data on "tube formation" is not useful - in any case, the cells look dead in the controls. Instead, move essential data from the supplement into the main paper such as 1) Suppl Fig. 5 on the PlexinB1 KO which should be complemented with data on the effect of the CRISPR/Cas lentivirus, 2) Suppl Fig. 7 on the characteristics of Src, VE-cad and FAK signaling in the STZ-OIR. Here the methods need to be more detailed than the current "Total protein of cells and retinas were extracted with a RIPA lysis buffer (end of page 3 in methods). 3) Suppl Fig 9 on the effect of knocking down PlexinB1.

5. The authors measure leakage of Evans blue injected in the tailvein and obtain remarkably consistent leakage effects in STZ-treated mice (see e.g. Fig. 3F-G, comparing WT and SEMA4D KO mice). In the various models/treatments, have the authors normalized for the difference e.g. in formation of angiogenic tufts. That is, do vessels leak once they have formed tufts in e.g. the absence of SEMA4D or PlexinB1, or is leakage reduced in the vessels that do form? To rephrase, is the effect of the treatments on the degree of pathological neovascularization or truly on restricting leakage from existing vessels?

Minor

6. The authors should discuss the paper by Bulloj et al., *Mol Neurobiol* 2018 May;55(5):4320-4332. Semaphorin4D-PlexinB1 Signaling Attenuates Photoreceptor Outer Segment Phagocytosis by Reducing Rac1 Activity of RPE Cells. Could some of the findings in the present study be explained based on the model presented by Bulloj?

7. The authors use an Abcam antibody against pVE-cadherin. Please specify which phosphorylation site the antibody is raised against.

8. The authors assess VE-cadherin internalization in sparse cells (Fig. 5). Please redo on dense cells with mature junctions.

9. Formation of acellular capillaries (Fig. 3H) seems to reflect vessel stability and would be better addressed by staining for collagen IV and IsolectinB4.

10. The authors show that PlexinB1 forms a complex with Src and that SEMA4D dramatically regulates accumulation of pSrc. Do the authors believe that PlexinB1 binds Src directly or indirectly via MET? This should be addressed.

11. The authors wish to stress a role for SEMA4D in regulation of VE-cadherin turnover and thereby in regulation of the vascular barrier. SEMA4D has a well-documented role in regulation of the actin cytoskeleton which is an important aspect of the junctional barrier. This should be addressed.

Referee #2 (Remarks for Author):

The manuscript by Wu et al explores the instrumental role of Semaphorins in retinal vascular pathologies; they report both convincing molecular mechanisms and in vivo insights, and also provide proof-of-concept of therapeutic uses.

1- What is the impact of Sema4D/PlxB1 on N-cadherin in endothelial cells? In line with this, is N-cadherin and VE-cadherin expression regulated in endothelial cells and pericytes?

2- The authors report modulation of FAK phosphorylation, but did not investigate its specific involvement in Sema4D-modulated permeability and angiogenic features?

3- All methods are available in the supplemental material, which is unusual - please indicate phospho-site for antibodies, this is critical information. How CRISPR cell lines were generated, and characterized required detailed information.

4- Besides STZ-induced retinopathies, retinal vascular permeability is also instrumental in age-related macular degeneration. It would be interesting to test Sema4D therapeutic strategies in artificial models of AMD, such as laser-induced macular degeneration.

5- Multiple labs have previously shown that Sema3A contributes to retinal angiogenesis and permeability. Here, the authors report sema3A up-regulation in their model. Could the authors evaluate in vitro the specific contribution of Sema3A, alone and in combination with Sema4D?

Referee #3 (Comments on Novelty/Model System for Author):

Find nothing problematic.

Referee #3 (Remarks for Author):

In the present study, Wu et al studied the role of Sema4D/PlexinB1 in the progression of diabetic retinopathy (DR) using mouse models and human samples. The data show Sema4D acts as a proangiogenic and destabilizing molecule *in vivo*. *In vitro* studies indicate this effect might be mediated by mDia1-Src pathway in endothelial cells and pericytes. Anti-Sema4D has an additive effect on anti-VEGF ameliorating the pathogenesis of DR models, suggesting a clinical utility as a combination therapy. In this reviewer's opinion, although a number of *in vitro* experiments support the authors' claim, *in vivo* data are not rigorous enough to support the conclusion, in particular all the genetic approaches are not cell type-specific. The effect on neurons could not be ignored given that these molecules are primarily known as repulsive guidance cues for neurons. In addition, the authors have to do some critical experiments before publication as listed below:

(Major)

1. (Fig. 1E) To precisely evaluate the source of Sema4D, *in situ* hybridization is required, as this molecule indeed exists as a soluble form.
2. (Fig. 3J; Fig. 8E; Fig. S10F) A decline in the desmin intensity in STZ-treated mice looks like morphological changes like retraction of processes, but not a reduction in the cell number or dropout. Staining with other markers like NG2 or PDGFRb together with nuclear staining might be helpful.
3. (Fig. 7E) Many previous reports indicate the VEGF/VEGFR inhibition attenuates not only pathologic angiogenesis but also physiological revascularization. However, it's not the case with this figure. Intra-ocular injection itself easily cause this kind of phenotype in OIR model.
4. (Fig. 8I; Fig. S8C) N-cadherin staining is not convincing at all. That should be membranous, but looks like a non-specific staining of some serum components?
5. (Fig. S8) These high magnification views of (cultured?) single cells are not convincing to show PlexinB1 is indeed expressed in endothelial cells and pericytes. *In situ* hybridization for PlexinB1 in the OIR retina should be shown.

(Minor)

2. (p. 3, line3) It should be mentioned that the gene for Sema4D contains upstream hypoxia response elements (HRE).

Additional correspondence

31 January 2019

AUTHOR COMMENTS:

Thank you very much for offering us the opportunity of revising our manuscript (EMM-2018-10154). Also thank you for your kindness in the extension of the revision time to 6 months. We will try our best to address all concerns raised by you and the reviewers.

We are writing to ask for your guidance on your notes regarding "referee 3 insists on using cell-specific conditional knockout mice, which was not addressed in your current rebuttal letter". The referee 3 commented that the genetic approaches were not cell type-specific and the effect on neurons could not be ignored *in vivo*. We are very sorry for the overlook in the rebuttal letter. Over the past few days, we have searched related publications for the floxed mice in order to establish cell type specific knockout mice; however, we found no indication of established cell-specific conditional knockout mice of Sema4D/PlexinB1 signaling so far. We have also consulted many mouse genetics companies for the possibility of generating new cell-specific conditional knockout mice, but were told that the generation of the floxed mice and breeding to cell-specific Cre mice will take more than one year not to mention characterization of the phenotypes. We are concerned about

the time-consuming process of cell-specific conditional knockout mice that will exceed the given revision time and may affect the novelty of our research for diabetic retinopathy (DR) therapy. One solution is, however, to use adeno-associated virus (AAV) carrying cell-specific promoter to drive expression of shRNA for knocking down Sema4D/PlexinB1 signaling in vivo. This method was used by recent high-quality publications [1, 2]. We can use an endothelial-specific promoter, ICAM-2 [3-5], to drive expression of shRNA in endothelial cells and a pericyte-specific promoter, PDGFR β [6], to drive expression of shRNA in pericytes in vivo. In this way, we can achieve the cell type-specific knockdown. Moreover, AAV vector mediated gene therapy has been validated by European Medicines Agency and also approved by the FDA in the USA for treating inherited retinal disease [7]. We believe that this method can also increase the translational medical implications of our findings. In addition, we will add experimental data to evaluate the effects of our treatments on neuron function.

Again, we appreciate the opportunity for the revision and would like to ask you for your advice on whether we could choose AAV mediated cell-specific knock-down to address this question instead of conditional knockout mice as suggested. Thank you very much for your attention and time. Looking forward to hearing from you.

1. Niu, C., et al., Metformin alleviates hyperglycemia-induced endothelial impairment by downregulating autophagy via the Hedgehog pathway. *Autophagy*, 2019: p. 1-28.
2. Hu, J., et al., Inhibition of soluble epoxide hydrolase prevents diabetic retinopathy. *Nature*, 2017. 552(7684): p. 248-252.
3. Huang, X., et al., Genome editing abrogates angiogenesis in vivo. *Nat Commun*, 2017. 8(1): p. 112.
4. Wu, W., et al., AAV-CRISPR/Cas9-Mediated Depletion of VEGFR2 Blocks Angiogenesis In Vitro. *Invest Ophthalmol Vis Sci*, 2017. 58(14): p. 6082-6090.
5. Dai, C., R.E. McAninch, and R.E. Sutton, Identification of synthetic endothelial cell-specific promoters by use of a high-throughput screen. *J Virol*, 2004. 78(12): p. 6209-21.
6. Armulik, A., G. Genove, and C. Betsholtz, Pericytes: developmental, physiological, and pathological perspectives, problems, and promises. *Dev Cell*, 2011. 21(2): p. 193-215.
7. FDA Advisory Committee Unanimously Recommends Approval of Investigational LUXTURN[®] (voretigene neparvovec) for Patients with Biallelic RPE65-mediated Inherited Retinal Disease. Available from: <https://globenewswire.com/news-release/2017/10/12/1145222/0/en/FDA-Advisory-Committee-Unanimously-Recommends-Approval-of-Investigational-LUXTURN-voretigene-neparvovec-for-Patients-with-Biallelic-RPE65-mediated-Inherited-Retinal-Disease.html>.

EDITOR REPLY:

Thank you for your email. I have gone back to the reviewer to ask about AAV cell-specific knock-down of Sema4D/Plexin B1, and here is what this reviewer said:

"... this approach is far less rigorous than canonical conditional knockout. [...] as far as I know there are no report showing EC-specific knockdown using AAV which is necessary for this study (i.e. EC or PC-specific KD of PlexinB1). Overall, this proposal is not impossible but is quite challenging. Rigorous monitoring of gene delivery specificity and efficiency is required."

I hope this helps and answers your concerns.

1st Revision - authors' response

23 May 2019

Referee #1:

In this study, Bo Hu and colleagues follow up on their previous data on SEMA4D in regulation of the BBB and that inhibition of SEMA4D improves the outcome of stroke in the rat (Faseb J 2018). They use a wide range of methods and models to address the role of SEMA4D in retinopathies. While some of the presented data (cleavage of SEMA4D, the role of PlexinB1 as a receptor for SEMA4D) have been presented before, there are novel data in this presentation. Of particular interest is the increased production of soluble SEMA4D in the aqueous fluid of patients with diabetic retinopathy. However, there are a number of serious concerns with this study, as listed below.

1. The variability of the data is unexpectedly small. Throughout, error bars are miniscule which calls into question whether all data points have been included and how data was processed. See for example i) the lentivirus-mediated knockdown of PlexinB1 in the retina followed by STZ-treatment for 3 months and blotting in Supplemental Figure 9 A and G. How can lentivirus-mediated delivery of CRISPR-Cas9 to the vitreous of individual mice, followed by STZ-treatment, again of individual mice, give rise to such all or none effect with very little variation?!

Response: We have included all the data points for reviewers to understand our results more clearly. The mean \pm SD value in CRISPR-pIB1 group of previous Supplemental Figure 9A is 0.5141 ± 0.1837 (The minimum value is 0.2464 and the maximum value is 0.721). We think that the variation is in normal range. The western blots are representative images of independent biological replicates. Please see detail description below (in response to question iv). Meanwhile, due to the non-specific PlexinB1 knockdown by CRISPR-pIB1, we added cell type-specific knockdown of PlexinB1 in vivo to replace these results (Fig 6).

Moreover, ii) how come that endothelial cells do not seem to grow to fill the wound at all in the control, in 24 h (see for example Fig. 4). In 24h, such dense cultures would be expected to completely fill the wound.

Response: If the endothelial cells were cultured in medium 131, such dense cultures would be expected to completely fill the wound. However, in our study, all confluent endothelial cells were starved in DMEM with 0.5% FBS overnight to make the cells quiescent before scratching. After scratching, the cells were still cultured in DMEM with 0.5% FBS containing low concentration of mitomycin-C during the wound repair according to the previous studies[1, 2] (mentioned in “Wound closure assays” section), so the cell migration is very little in the control. We described it clearer in revised manuscript.

iii) How come pSrc/pFAK levels are barely detected in tissue culture cells before they are treated with e.g. Sema4D (see Fig. 5B). Adherent cells in tissue culture have high levels of pSrc/pFAK through integrin-dependent activation. How do these cells remain attached at all?

Response: We can see from the Figure A below, the pSrc/pFAK intensity are strong when the cells are cultured in the medium 131, but they are barely detected in the DMEM medium with 0.5% FBS. In our study, the cells for phosphorylation detection were starved in DMEM with 0.5% FBS before different stimulations. The plates were pre-coated with gelatin so that the culture cells could remain attached after long time starvation. We included these experimental details in our revised manuscript (“Western blot analysis” and “Cell culture” section). Moreover, previous studies also showed bare pSrc/pFAK levels in non-stimulated adherent endothelial cells[3-5] (see figure B below, cited from figure 3 of reference 3).

iv) The retinopathy field agrees it is challenging to detect vessel leakage from the retinal superficial vasculature. Still the authors manage to obtain 10-fold induction of leakage (see Fig. 3F-G). For the

in vivo analyses, the authors state throughout that the n=6; I assume this means 6 mice or is it 3 mice, using each of the two eyes/mouse as individual samples (which should not be done)? The authors need to declare whether these were selected from a much larger cohort. How many times did they do individual repeats (should be 3 times). How do they explain to have a variability of less than 10% in these complicated in vivo analyses? In their present form, the results are not credible.

Response: We fully agree with the reviewer that it is challenging to detect vessel leakage from normal retinal vasculature. However, previous studies[6-8] confirmed that the leakage was evident induction in STZ-treated mice (see figure C below, cited from figure 5 of reference 6). We increased the samples in Figure 3G and our results showed similar induction with previous studies. All n values represented the number of independent biological replicates in our study. For in vivo analyses, we indicated that the n values were range from 5-13. For PCR in vivo, two same-treated retinas were mixed as one independent biological replicate for RNA extraction. For Western blotting in vivo, four same-treated retinas were mixed as one independent biological replicate for protein extraction. For PCR, Western blotting, ELISA and Evans blue assays, one data point represented the mean value of three technical replicates from one independent biological replicate. The pericytes coverage percentage, VE-cadherin continuity percentage and N-cadherin coverage percentage per retina were the mean values from ten random fields. For two same-treated retinas of one mouse, the average value of two retinas was indicated as one data point.

All animals within each experiment were from the same strain, similar age (within the week), same gender, same fur colour, similar weight and comparable health status. The animals were randomized into different groups from a much larger cohort. For OIR model, we did IsolectinB4 staining for each batch of mice to determine if the models are successful (those failed batches were discarded). We also increased the mice number per group (n=10-13) for neovascularization analysis in OIR model to make the data more credible. For STZ mice, we continuously monitored fasting blood glucose per month to ensure the models were successful (those failed mice were excluded).

The mice received intravitreal injections were treated with antibiotics to prevent infection. Those with eyeball infections after injection or with abnormal conditions due to other reasons were excluded in OIR and STZ model. We included these details in the revised manuscript. For each experiment, we did a lot of training to stabilize the experimental operation before formal experiments. All these efforts were made to reduce the variability in vivo.

C

cited from figure 5 of reference 6

2. The authors suggest that anti-SEMA4D should be used together with anti-VEGF antibodies to treat DR, based on that SEMA4D KO mice show more limited retinal damages and that treatment of STZ-induced DR with anti-SEMA4D antibodies is beneficial. In Fig. 1 D, they show that there are elevated levels of SEMA4D in aqueous fluid of controls and DR patients, treated with anti-VEGF antibodies. Fig. 1E and F show that there is less soluble SEMA4D in patients with a more severe disease (with higher CST and MV change). The rationale for treating patients with a more severe disease with anti-SEMA4D antibodies therefore does not seem to exist, as these patients based on the cohort shown in Fig. 1E and F, have very low levels of SEMA4D. Please explain the rationale better.

Response: We are sorry that we have not described clearly about these panels. In Fig. 1 D, the levels of sSEMA4D in aqueous fluid of DR patients were measured at baseline before initial anti-VEGF therapy. Central subfield thickness (CST) and macular volume (MV) were measured by OCT at baseline before initial anti-VEGF therapy and 3 months after initial anti-VEGF therapy. The CST and MV change were the baseline value minus the 3 months value. The higher CST and MV change mean the patients are more sensitive to anti-VEGF therapy. We found that higher levels of baseline sSEMA4D associated with less CST and MV change (less response to anti-VEGF therapy). This indicated that sSema4D levels in aqueous fluid may help early identification of non- or poor responders of current anti-VEGF therapy. Our other results also suggested that anti-SEMA4D may improve anti-VEGF treatment in DR. We described it clearer in our revised manuscript.

3. For knocking down PlexinB1, the authors perform intravitreal injection of lentivirus (1 μ l) expressing CRISPR-wt or CRISPR-pIB1 at P7. Please give exact information on the degree of PlexinB1 knockdown in retinal endothelial cells by PCR and by immunostaining. PlexinB1 is broadly expressed in general. Show by immunostaining which cells in the retina express PlexinB1 before and after CRISPR-pIB1. Discuss how the biology still can be attributed to knockdown of PlexinB1 in endothelial cells.

Response: We appreciate the reviewer's valuable comments. Due to the non-specific PlexinB1 knockdown by CRISPR-pIB1, we added cell type-specific approaches *in vivo* to strengthen our conclusion. We used Tie2-Cre mice to specific express Cre in endothelial cells. Adenoviruses containing double-floxed Cre-inducible GFP and mir30-shRNA targeted against PlexinB1 were intravitreally injected into these mice. The GFP staining showed that the shRNA was mainly expressed in endothelial cells. Western blotting confirmed knockdown of PlexinB1 in retinas. Our results indicated that knockdown of PlexinB1 in endothelial cells reduced the pathologic retinal neovascularization and vascular leakage in OIR model (Fig 6).

4. The study comprises a lot of data; in addition to the main figure, 11 supplemental figures with many panels each, are provided. The impression is of brute force rather than assembling the presentation around a theme. The methods are very sparsely described. Some methods/results do not need to be included such as the data on Sema4D cleavage or regulation by IRF1. The data on "tube formation" is not useful - in any case, the cells look dead in the controls. Instead, move essential data from the supplement into the main paper such as 1) Suppl Fig. 5 on the PlexinB1 KO which should be complemented with data on the effect of the CRISPR/Cas lentivirus, 2) Suppl Fig. 7 on the characteristics of Src, VE-cad and FAK signaling in the STZ-OIR. Here the methods need to be more detailed than the current "Total protein of cells and retinas were extracted with a RIPA lysis buffer (end of page 3 in methods). 3) Suppl Fig 9 on the effect of knocking down PlexinB1.

Response: We appreciate the reviewer's valuable comments and made extensive rearrangement of our manuscript. First, we added some detailed methodological descriptions. Second, the results on Sema4D cleavage and regulation by IRF1 could partly explain how Sema4D was increased in DR. Meanwhile, the expression pattern of IRF1 and AMAD17 in our disease context hadn't been explored in previous study, so we deleted some data and kept some key results. Third, the tube formation assays were done in the Matrigel, the Matrigel is three-dimensional after solidification in the culture plate, the cells may grow in three-dimensional. Since not in a plane, the cells were black or bright when taking pictures with a microscope. In addition, the parameters of the microscope may not be adjusted to the optimal state, so we deleted all data about tube formation. Fourth, the previous results of Suppl Figure5 were now included into Figure5. Previous Suppl Figure7 and Suppl Figure8 were integrated into Expanded FigureEV3 in order to improve their accessibility for readers. We added more details about the protein extraction of retinas (see "Western blot analysis" section). We implemented cell type-specific approaches to knockdown PlexinB1 in endothelial cells and pericytes respectively *in vivo*. The results were shown in Figure 6.

5. The authors measure leakage of Evans blue injected in the tailvein and obtain remarkably consistent leakage effects in STZ-treated mice (see e.g. Fig. 3F-G, comparing WT and SEMA4D KO mice). In the various models/treatments, have the authors normalized for the difference e.g. in formation of angiogenic tufts. That is, do vessels leak once they have formed tufts in e.g. the absence of SEMA4D or PlexinB1, or is leakage reduced in the vessels that do form? To rephrase, is the effect of the treatments on the degree of pathological neovascularization or truly on restricting leakage from existing vessels?

Response: We are very grateful for the reviewer's in-depth considerations. Indeed, our study provided evidence to distinguish the respective therapeutic effects of anti-Sema4D on pathological neovascularization and vascular leakage. It is well known that STZ-induced DR model develops only retinal vascular leakage but not pathological neovascularization and is used to mimic clinical diabetic macular edema (DME). The OIR model induces both pathological neovascularization and vascular leakage, the vascular leakage in OIR model is largely depend on pathological neovascularization. The OIR model is used to mimic clinical proliferative diabetic retinopathy (PDR). The results from STZ model (Fig 3G, 6K, 7M, 8B, 8L) indicated that inhibition of Sema4D/PlexinB1 signaling restricted leakage from existing vessels. The results from OIR model (Fig 3A-3E, 6B-6F, 7A-7I) indicated that inhibition of Sema4D/PlexinB1 signaling reduced pathological neovascularization and vascular leakage (the leakage was an accompanying consequence, we just checked by the way in OIR model). Our study suggested that inhibition of Sema4D/PlexinB1 signaling may provide benefits for either DME or PDR (the two main manifestations of diabetic retinopathy that require treatment).

Moreover, it is common that many patients have both DME and PDR in clinical practice, they have both leakage from existing vessels and leakage from pathological neovascularization. For clinicians, we are not going to distinguish them, but looking for ways to treat both leakage and neovascularization. So our findings is important for therapeutic uses.

Minor

6. The authors should discuss the paper by Bulloj et al., Mol Neurobiol 2018 May;55(5):4320-4332. Semaphorin4D-PlexinB1 Signaling Attenuates Photoreceptor Outer Segment Phagocytosis by Reducing Rac1 Activity of RPE Cells. Could some of the findings in the present study be explained based on the model presented by Bulloj?

Response: The paper detected that Sema4D/PlexinB1 signaling had a novel physiological function in photoreceptor outer segment phagocytosis by retinal pigment epithelium. The process is essential for the survival and function of photoreceptors in the retina. So it reminds us that comprehensive evaluations of Sema4D knockout and anti-Sema4D on retinal neuronal function are needed in the future. Related research is in progress in our lab and the results will be reported in our next study. We discussed this in our revised manuscript.

7. The authors use an Abcam antibody against pVE-cadherin. Please specify which phosphorylation site the antibody is raised against.

Response: The phosphorylation site was Y685 (Abcam, ab119785) according to previous studies [9-11]. We added this information in revised manuscript.

8. The authors assess VE-cadherin internalization in sparse cells (Fig. 5). Please redo on dense cells with mature junctions.

Response: We redid the experiments in vitro as suggested (Fig 4K in the revised manuscript).

9. Formation of acellular capillaries (Fig. 3H) seems to reflect vessel stability and would be better addressed by staining for collagen IV and IsolectinB4.

Response: We redid the staining with collagen IV and IsolectinB4 as suggested. The results indicated a same conclusion (Fig 3H and 3I). Periodic acid-Schiff staining after retinal trypsin digestion is often used to show formation of acellular capillaries during diabetic retinopathy[6, 12-15], so we still kept the relevant data in the revised manuscript.

10. The authors show that PlexinB1 forms a complex with Src and that SEMA4D dramatically regulates accumulation of pSrc. Do the authors believe that PlexinB1 binds Src directly or indirectly via MET? This should be addressed.

Response: We used specific siRNA to silence MET expression in endothelial cells and found that silencing MET didn't affect the formation of PlexinB1/Src complex (see figure below). Previous studies reported that PlexinB1 could transmit its signaling through MET dependent and independent ways under different conditions[16, 17]. Meanwhile, PlexinB1 could activate or suppress MET in different diseases[16, 18]. So future comprehensive evaluations are needed to explore its exact role.

11. The authors wish to stress a role for SEMA4D in regulation of VE-cadherin turnover and thereby in regulation of the vascular barrier. SEMA4D has a well-documented role in regulation of the actin cytoskeleton which is an important aspect of the junctional barrier. This should be addressed.

Response: We appreciate the reviewer's valuable comments. The adherens junctions and tight junctions of endothelial cell both play important roles in the maintenance of vascular integrity[19, 20]. VE-cadherin is a component of endothelial cell-to-cell adherens junctions. In the present study, we identified that Sema4D/PlexinB1 induced endothelial cell VE-cadherin dysfunction through mDIA1-Src pathway. Meanwhile, Sema4D knockout and anti-Sema4D could ameliorate VE-cadherin dysfunction in DR model. However, previous studies reported that SEMA4D had a well-documented role in regulation of actin cytoskeleton[21, 22]. It is known that adherens junctions and tight junctions have complicated interactions with actin cytoskeleton[19, 20]. It's possible that Sema4D may also induce VE-cadherin or other junction dysfunction through regulation of the actin cytoskeleton in our disease context, and this needs to be further examined. We discussed this issue in our revised manuscript.

Referee #2:

The manuscript by Wu et al explores the instrumental role of Semaphorins in retinal vascular pathologies; they report both convincing molecular mechanisms and in vivo insights, and also provide proof-of-concept of therapeutic uses. 1- What is the impact of Sema4D/PlxB1 on N-cadherin in endothelial cells? In line with this, is N-cadherin and VE-cadherin expression regulated in endothelial cells and pericytes?

Response: Our result indicated that pericytes expressed much more N-cadherin than endothelial cells (see figure below), so we mainly explored the impact of Sema4D on N-cadherin internalization in pericytes. In addition, we tested the N-cadherin and VE-cadherin expression in endothelial cells/pericytes co-culture as suggested. The results showed that Sema4D decreased both the N-cadherin and VE-cadherin expression in the co-culture system (Appendix Fig S6E).

2- The authors report modulation of FAK phosphorylation, but did not investigate its specific involvement in Sema4D-modulated permeability and angiogenic features?

Response: We used a selective FAK inhibitor (GSK2256098) to investigate the role of FAK phosphorylation in Sema4D signaling. The results indicated that the FAK inhibitor partly blocked Sema4D-induced endothelial cell migration and permeability (Appendix Fig S4F - H).

3- All methods are available in the supplemental material, which is unusual - please indicate phospho-site for antibodies, this is critical information. How CRISPR cell lines were generated, and characterized required detailed information.

Response: Because it was a first submission, we didn't reformat the manuscript. We reformatted it in our revised manuscript. The phospho-sites were p-FAK (Tyr576/577), p-Src antibody (Tyr416), p-VE-cadherin (Y685) (added in the "Western blot analysis" section). The CRISPR cell lines were generated by lentivirus containing single guide RNA (sgRNA) and Cas9. Briefly, the online CRISPR design tool (<http://crispr.mit.edu>) was used to predict the sgRNA sequences for the mouse PlexinB1 gene. The targeting sequence used in this study was 5'- GTCGCCGGTCGGAGTGCTCA -3'. DNA oligos were synthesized and cloned into vectors containing CAS9. Then the vectors were packaged into lentivirus. After purification of the CRISPR/Cas9 lentivirus, the lentivirus was transfected into the targeted cells according to the manufacturer's instructions. The efficiency was determined by Western blot analysis. (added in the "Viral vectors" section). We added these details in the revised manuscript.

4- Besides STZ-induced retinopathies, retinal vascular permeability is also instrumental in age-related macular degeneration. It would be interesting to test Sema4D therapeutic strategies in artificial models of AMD, such as laser-induced macular degeneration.

Response: We fully agree with the reviewer that it will be interesting to test whether anti-Sema4D could be used as a treatment for AMD. However, our clinical samples and mouse models in this study were all for diabetic retinopathy. We think the idea may be more appropriate to be solved in our future research, so we had discussed this in the manuscript (in the last paragraph of the discussion section).

5- Multiple labs have previously shown that Sema3A contributes to retinal angiogenesis and permeability. Here, the authors report sema3A up-regulation in their model. Could the authors evaluate in vitro the specific contribution of Sema3A, alone and in combination with Sema4D?

Response: We evaluated the effect in vitro as suggested. We found that Sema3A (500ng/ml) inhibited endothelial cell migration but induced endothelial permeability (see figure below). Meanwhile, Sema3A had a synergistic effect with Sema4D on endothelial permeability.

Previous study reported that Sema3A was induced in early phases of STZ model within the neuronal retina to trigger vascular leakage[3]. However, its role in pathologic retinal neovascularization varied[23-25]. Two studies indicated that Sema3A formed a repulsive barrier to misdirect endothelial cells toward the vitreous[23] and acted as a potent attractant for mononuclear phagocytes [25] to induce pathological preretinal neovascularization. Another study suggested that Sema3A could inhibit angiogenic features of endothelial cells to reduce neovascularization in OIR model[24]. Therefore, semaphorins may play different roles as DR progresses. Whether synergistic regulation or precise intervention of different semaphorins (Sema4D and Sema3A) according to the

disease stage provide more benefits needs to be studied in the future. We discussed this issue in our revised manuscript

Referee #3:

In the present study, Wu et al studied the role of Sema4D/PlexinB1 in the progression of diabetic retinopathy (DR) using mouse models and human samples. The data show Sema4D acts as a proangiogenic and destabilizing molecule *in vivo*. *In vitro* studies indicate this effect might be mediated by mDia1-Src pathway in endothelial cells and pericytes. Anti-Sema4D has an additive effect on anti-VEGF ameliorating the pathogenesis of DR models, suggesting a clinical utility as a combination therapy. In this reviewer's opinion, although a number of *in vitro* experiments support the authors' claim, *in vivo* data are not rigorous enough to support the conclusion, in particular all the genetic approaches are not cell type-specific. The effect on neurons could not be ignored given that these molecules are primarily known as repulsive guidance cues for neurons. In addition, the authors have to do some critical experiments before publication as listed below:

Response: As suggested, we added cell type-specific approaches *in vivo* to strengthen our conclusion. We used Tie2-Cre and PDGFR β -Cre mice to specifically express Cre recombinase in endothelial cells and pericytes respectively. Adenoviruses containing a potent expression of PlexinB1 shRNA that was activated by Cre recombinase-mediated recombination were intravitreally injected into these mice [26-28]. The GFP staining showed that the shRNA was mainly expressed in endothelial cells and pericytes respectively. Western blotting confirmed that PlexinB1 was knocked down in retinas. Our results indicated that knockdown of PlexinB1 in endothelial cells reduced the pathologic retinal neovascularization and vascular leakage in OIR model. Meanwhile, knockdown of PlexinB1 in endothelial cells or pericytes, respectively, alleviated STZ-induced vascular leakage (Fig 6).

Retinal neuronal dysfunction is also one aspect of diabetic retinopathy [29-32]. Our results also showed that Sema4D knockout reduced the TUNEL positive retinal ganglion cell (see figure below). However, comprehensive evaluations of Sema4D knockout and anti-Sema4D on retinal neuronal function is in progress in our lab. So these results will be reported in our next study. We discussed the issue in the revised manuscript.

[Unpublished data removed at the authors' request]

(Major)

1. (Fig. 1E) To precisely evaluate the source of Sema4D, *in situ* hybridization is required, as this molecule indeed exists as a soluble form.

Response: Fluorescence *in situ* hybridization also indicated that Sema4D mRNA is evidently localized in GFAP⁺ glial cells in OIR retinas (Fig 2F).

2. (Fig. 3J; Fig. 8E; Fig. S10F) A decline in the desmin intensity in STZ-treated mice looks like morphological changes like retraction of processes, but not a reduction in the cell number or

dropout. Staining with other markers like NG2 or PDGFRb together with nuclear staining might be helpful.

Response: We are sorry for the vague descriptions of the pericytes coverage value in the Figures, the pericytes coverage is the desmin positive area divided by collagen IV positive area (added in “Immunofluorescence” section). The value in the Figures was fold change compared to control. The method, combining pericytes staining (NG2, PDGFR β , DESMIN) with vascular staining (CD31, Collagen IV, isolectin B4) to calculate pericytes coverage, had been used by many previous studies[15, 33-35] to partly represent pericyte-associated pathological changes. As suggested by the reviewer, we stained PDGFR β together with nuclear staining using retinal whole mount to strengthen the conclusion. We found that Sema4D knockout and anti-Sema4D blocked pericyte-associated pathological changes both in terms of cell count and cell coverage (Fig 3J-3L, 8E, 8F).

3. (Fig. 7E) Many previous reports indicate the VEGF/VEGFR inhibition attenuates not only pathologic angiogenesis but also physiological revascularization. However, it's not the case with this figure. Intra-ocular injection itself easily cause this kind of phenotype in OIR model.

Response: We appreciate the reviewer's valuable comments. It is well known that VEGF/VEGFR inhibition indeed attenuates retinal vascular outgrowth during development. However, previous reports about the effects of VEGF/VEGFR inhibition in physiological revascularization in OIR model were controversial. Several studies[36-38] suggested that VEGF/VEGFR inhibition attenuated not only pathologic angiogenesis but also vaso-obliteration area, while another study[39] found that VEGF/VEGFR inhibition didn't affect vaso-obliteration area. In order to avoid confusion for the reader, we didn't evaluate this aspect in our OIR model. Meanwhile, we had used intravitreal injections of PBS or IgG to rule out the impact of intra-ocular injection itself on the results.

4. (Fig. 8I; Fig. S8C) N-cadherin staining is not convincing at all. That should be membranous, but looks like a non-specific staining of some serum components?

Response: We thank for the reviewer's careful observation. N-cadherin is a membrane protein but its distribution is irregular. For N-cadherin staining, we had tested antibodies from different companies (CST, Abcam, R&D systems, Life Technologies) and found that the antibody from Life Technologies was the best. We had used negative controls to exclude non-specific staining during our experiments (see Figure D below). A previous study from *Nature* Journal also showed similar staining results[15] (see figure E below, cited from figure 2 of reference 15).

cited from figure 2 of reference 15

5. (Fig. S8) These high magnification views of (cultured?) single cells are not convincing to show PlexinB1 is indeed expressed in endothelial cells and pericytes. In situ hybridization for PlexinB1 in the OIR retina should be shown.

Response: We appreciate the reviewer's valuable comments. We used probes from different companies and repeatedly optimized the experimental conditions, however, the hybridization stainings for PlexinB1 were still very poor. In addition, because endothelial cells and pericytes are located close to each other, staining is difficult to accurately distinguish the expression of PlexinB1 in each cell. As an alternative, we used fluorescence-activated cell sorting (FACS) with antibodies against CD31 (endothelial cells) and CD13 (pericytes) to isolate CD31⁺ CD13⁻ endothelial cells and CD31⁻ CD13⁺ pericytes in retinas, according to previous studies[40, 41]. Then we ran PCR to detect the mRNA levels of PlexinB1 in the fresh sorted cells (Fig 5C, S5A). The results indicated that PlexinB1 was expressed in endothelial cells and pericytes. Meanwhile, our previous data detected the protein expression of PlexinB1 in cultured endothelial cells and pericytes (Fig 5D). Flow analysis showed that the purity was 97.4% for cultured endothelial cells and was 97.5% for cultured pericytes (Fig S5B). We thought these results indicated PlexinB1 was expressed in endothelial cells and pericytes to a certain extent.

(Minor)

2. (p. 3, line3) It should be mentioned that the gene for Sema4D contains upstream hypoxia response elements (HRE).

Response: we discussed it in our revised manuscript.

References

- Zhang, X.P., et al., *Ginsenoside Rh2 inhibits vascular endothelial growth factor-induced corneal neovascularization*. *FASEB J*, 2018. **32**(7): p. 3782-3791.
- Tang, H., et al., *siRNA-knockdown of ADAMTS-13 modulates endothelial cell angiogenesis*. *Microvasc Res*, 2017. **113**: p. 65-70.
- Cerani, A., et al., *Neuron-derived semaphorin 3A is an early inducer of vascular permeability in diabetic retinopathy via neuropilin-1*. *Cell Metab*, 2013. **18**(4): p. 505-18.
- Lee, C.S., et al., *Dipeptidyl Peptidase-4 Inhibitor Increases Vascular Leakage in Retina through VE-cadherin Phosphorylation*. *Sci Rep*, 2016. **6**: p. 29393.
- Miloudi, K., et al., *Truncated netrin-1 contributes to pathological vascular permeability in diabetic retinopathy*. *J Clin Invest*, 2016. **126**(8): p. 3006-22.
- Shan, K., et al., *Circular Noncoding RNA HIPK3 Mediates Retinal Vascular Dysfunction in Diabetes Mellitus*. *Circulation*, 2017. **136**(17): p. 1629-1642.
- Liu, C., et al., *Targeting pericyte-endothelial cell crosstalk by circular RNA-cPWWP2A inhibition aggravates diabetes-induced microvascular dysfunction*. *Proc Natl Acad Sci U S A*, 2019. **116**(15): p. 7455-7464.
- Huang, H., et al., *Deletion of placental growth factor prevents diabetic retinopathy and is associated with Akt activation and HIF1alpha-VEGF pathway inhibition*. *Diabetes*, 2015. **64**(1): p. 200-12.
- Akla, N., et al., *BMP9 (Bone Morphogenetic Protein-9)/Alk1 (Activin-Like Kinase Receptor Type I) Signaling Prevents Hyperglycemia-Induced Vascular Permeability*. *Arterioscler Thromb Vasc Biol*, 2018. **38**(8): p. 1821-1836.
- Zhang, P., et al., *Mutant B-Raf(V600E) Promotes Melanoma Paracellular Transmigration*

- by *Inducing Thrombin-mediated Endothelial Junction Breakdown*. J Biol Chem, 2016. **291**(5): p. 2087-106.
11. Wessel, F., et al., *Leukocyte extravasation and vascular permeability are each controlled in vivo by different tyrosine residues of VE-cadherin*. Nat Immunol, 2014. **15**(3): p. 223-30.
 12. Yan, B., et al., *lncRNA-MIAT regulates microvascular dysfunction by functioning as a competing endogenous RNA*. Circ Res, 2015. **116**(7): p. 1143-56.
 13. Liu, C., et al., *Silencing Of Circular RNA-ZNF609 Ameliorates Vascular Endothelial Dysfunction*. Theranostics, 2017. **7**(11): p. 2863-2877.
 14. Lai, D.W., et al., *TPL2 (Therapeutic Targeting Tumor Progression Locus-2)/ATF4 (Activating Transcription Factor-4)/SDF1alpha (Chemokine Stromal Cell-Derived Factor-alpha) Axis Suppresses Diabetic Retinopathy*. Circ Res, 2017. **121**(6): p. e37-e52.
 15. Hu, J., et al., *Inhibition of soluble epoxide hydrolase prevents diabetic retinopathy*. Nature, 2017. **552**(7684): p. 248-252.
 16. Giordano, S., et al., *The semaphorin 4D receptor controls invasive growth by coupling with Met*. Nat Cell Biol, 2002. **4**(9): p. 720-4.
 17. Negishi-Koga, T., et al., *Suppression of bone formation by osteoclastic expression of semaphorin 4D*. Nat Med, 2011. **17**(11): p. 1473-80.
 18. Soong, J., et al., *Sema4D, the ligand for Plexin B1, suppresses c-Met activation and migration and promotes melanocyte survival and growth*. J Invest Dermatol, 2012. **132**(4): p. 1230-8.
 19. Hartsock, A. and W.J. Nelson, *Adherens and tight junctions: structure, function and connections to the actin cytoskeleton*. Biochim Biophys Acta, 2008. **1778**(3): p. 660-9.
 20. Giannotta, M., M. Trani, and E. Dejana, *VE-cadherin and endothelial adherens junctions: active guardians of vascular integrity*. Dev Cell, 2013. **26**(5): p. 441-54.
 21. Bulloj, A., et al., *Semaphorin4D-PlexinB1 Signaling Attenuates Photoreceptor Outer Segment Phagocytosis by Reducing Rac1 Activity of RPE Cells*. Mol Neurobiol, 2018. **55**(5): p. 4320-4332.
 22. Tasaka, G., M. Negishi, and I. Oinuma, *Semaphorin 4D/Plexin-B1-mediated M-Ras GAP activity regulates actin-based dendrite remodeling through Lamellipodin*. J Neurosci, 2012. **32**(24): p. 8293-305.
 23. Joyal, J.S., et al., *Ischemic neurons prevent vascular regeneration of neural tissue by secreting semaphorin 3A*. Blood, 2011. **117**(22): p. 6024-35.
 24. Yu, W., et al., *Inhibition of pathological retinal neovascularization by semaphorin 3A*. Mol Vis, 2013. **19**: p. 1397-405.
 25. Dejda, A., et al., *Neuropilin-1 mediates myeloid cell chemoattraction and influences retinal neuroimmune crosstalk*. J Clin Invest, 2014. **124**(11): p. 4807-22.
 26. Namikawa, K., et al., *A newly modified SCG10 promoter and Cre/loxP-mediated gene amplification system achieve highly specific neuronal expression in animal brains*. Gene Ther, 2006. **13**(16): p. 1244-50.
 27. Yokoo, T., et al., *Inflamed glomeruli-specific gene activation that uses recombinant adenovirus with the Cre/loxP system*. J Am Soc Nephrol, 2001. **12**(11): p. 2330-7.
 28. Kinoshita, K., et al., *Targeted and regulable expression of transgenes in hepatic stellate cells and myofibroblasts in culture and in vivo using an adenoviral Cre/loxP system to antagonise hepatic fibrosis*. Gut, 2007. **56**(3): p. 396-404.
 29. Duh, E.J., J.K. Sun, and A.W. Stitt, *Diabetic retinopathy: current understanding, mechanisms, and treatment strategies*. JCI Insight, 2017. **2**(14).
 30. Fu, Z., et al., *Fibroblast Growth Factor 21 Protects Photoreceptor Function in Type 1 Diabetic Mice*. Diabetes, 2018. **67**(5): p. 974-985.
 31. Zhang, X., et al., *Diabetic retinopathy: neuron protection as a therapeutic target*. Int J Biochem Cell Biol, 2013. **45**(7): p. 1525-9.
 32. Kern, T.S. and A.J. Barber, *Retinal ganglion cells in diabetes*. J Physiol, 2008. **586**(18): p. 4401-8.
 33. Park, D.Y., et al., *Plastic roles of pericytes in the blood-retinal barrier*. Nat Commun, 2017. **8**: p. 15296.
 34. Ogura, S., et al., *Sustained inflammation after pericyte depletion induces irreversible blood-retina barrier breakdown*. JCI Insight, 2017. **2**(3): p. e90905.
 35. Kim, J., et al., *YAP/TAZ regulates sprouting angiogenesis and vascular barrier maturation*. J Clin Invest, 2017. **127**(9): p. 3441-3461.
 36. Bucher, F., et al., *Antibody-Mediated Inhibition of Tspan12 Ameliorates Vasoproliferative Retinopathy Through Suppression of beta-Catenin Signaling*. Circulation, 2017. **136**(2): p.

- 180-195.
37. Michael, I.P., et al., *Local acting Sticky-trap inhibits vascular endothelial growth factor dependent pathological angiogenesis in the eye*. EMBO Mol Med, 2014. **6**(5): p. 604-23.
 38. Ridano, M.E., et al., *Galectin-1 expression imprints a neurovascular phenotype in proliferative retinopathies and delineates responses to anti-VEGF*. Oncotarget, 2017. **8**(20): p. 32505-32522.
 39. Oubaha, M., et al., *Senescence-associated secretory phenotype contributes to pathological angiogenesis in retinopathy*. Sci Transl Med, 2016. **8**(362): p. 362ra144.
 40. Crouch, E.E. and F. Doetsch, *FACS isolation of endothelial cells and pericytes from mouse brain microregions*. Nat Protoc, 2018. **13**(4): p. 738-751.
 41. Crouch, E.E., et al., *Regional and stage-specific effects of prospectively purified vascular cells on the adult V-SVZ neural stem cell lineage*. J Neurosci, 2015. **35**(11): p. 4528-39.

Additional correspondence - manuscript withdrawn by the authors

6 June 2019

AUTHOR COMMENTS:

We submitted a revised manuscript titled " Inhibition of Sema4D/PlexinB1 signaling alleviates vascular dysfunction in diabetic retinopathy " (manuscript number: EMM-2018-10154-V2). The manuscript is under peer review now. We would indeed very much like to publish in your Journal. However, a package containing our mobile hard drive and computer lost last week in the process of moving, the mobile hard drive and computer stored a lot of source data of this manuscript. We searched it for a week and found that the possibility of getting the package back was small. We are worried that the loss of source data will create potential risks for this manuscript. For this reason, we have to withdrawal this manuscript at present. We will redo the lost data and submit the manuscript after completing all the lost data. We apologize for any inconvenience caused by the termination of this submission. Once again, thanks very much for the time of you and reviewers.

EDITOR REPLY:

Thank you for contacting us.

We take note of your request to withdraw the manuscript. I will suspend the peer-review process until further notice and notify the referees.

Manuscript resubmitted

2 October 2019

We have repeated the experiments and obtained new data. We have got the same conclusion as the previous manuscript and would like to re-submit it for your consideration. We have addressed the comments raised by the reviewers.

Thank you very much for your attention and time. Looking forward to hearing from you.

2nd Editorial Decision

13 November 2019

Thank you for the submission of your revised manuscript to EMBO Molecular Medicine. We have now received the referees' reports, and as you will see they are supportive of publication of your study pending minor revisions. I am therefore pleased to inform you that we will be able to accept your manuscript once you've made the requested minor editorial changes, and have carefully addressed the concerns from referee #1 and #2 in the manuscript and in a point-by-point rebuttal.

I look forward to reading a new revised version of your manuscript as soon as possible.

2nd Revision - authors' response

2 December 2019

Referee #1 (Comments on Novelty/Model System for Author):

The authors have improved their study; there is a lot of data and challenging to evaluate. Although

I'm more convinced now than when I read the first submission, I'm still skeptical about some of the data for example the 10-fold induction of leakage in the retinal vessels. This is not what the expert laboratories report. There are more examples, such as the Src kinase inhibitor which binds tubulin; see my response to authors.

Response: We thank the review's comment about the improvement of our study. We are also touched by the rigorous scientific attitude of the reviewer. We summarized more literatures to answer the question about leakage in the retinal vessels. There were many high-quality publications manage to obtain 10-fold induction of retinal leakage by using Evans Blue absorbance measurement with STZ-induced mouse DR model (the same model and detection method with our study). Please see the cited figures: Figure5C from *Circulation*. 2017 Oct [1]; Figure2B and Figure5D from *Theranostics*. 2017 Jul [2]; Figure2D and FigureS3C from *Proc Natl Acad Sci U S A*. 2019 Apr [3]. A paper using Ins2^{Akita} mice, another diabetic retinopathy mouse model, obtained 4-fold induction of retinal leakage by quantifying FITC-BSA fluorescence signal (please see Figure2H from *Nature*. 2017 Dec [4]). However, a literature obtained only 2-fold induction of leakage by using Evans Blue absorbance measurement with STZ-induced rat DR model (please see Figure3B from *Circ Res*. 2015 Mar [5]). So the fold change of retinal leakage may vary with species, models and detection methods. For question about Src kinase inhibitor, please see response blue.

Referee #1 (Remarks for Author):

Wu et al., have performed an ambitious revision and argue well in their rebuttal.

Response: We sincerely thank the reviewer for providing insightful comments contributing to the improvement of the manuscript.

However, the drug denoted KX2-391 (now should be denoted Tirbanibulin) is not Src specific according to the literature, but inhibits also tubulin polymerization. See Niu et al., *JBC* Oct 2019. This should be amended.

Response: We appreciate the reviewer's valuable comments. First, the drug KX2-391 was purchased from Selleck Chemicals company (Houston, Texas, USA) (mentioned in "Cell culture and reagents" section). The chemical name of the drug is KX2-391 in the official website of Selleck Chemicals company. The Tirbanibulin seems to be the clinical product name of KX2-391. So we still use its chemical name of the drug (KX2-391) in the manuscript. Second, it is common that that many kinase inhibitors may inevitably have non-specific other targets. Other Src inhibitors may both target Src kinases and other kinases of Src family (such as Lck, Fyn, Lyn, Yes). KX2-391 is more specific to Src kinase among Src inhibitors and doesn't target other Src family kinases. Meanwhile, KX2-391 is the first clinical Src inhibitor with low toxicity. So we chose KX2-391 in our experiments. We are very grateful to the reviewer for pointing this out and hope that the reviewer will understand this unavoidable non-specificity. As suggested, we rewrote all sentence "a specific inhibitor of Src (KX2-391)" to "an inhibitor of Src (KX2-391)" in revised manuscript.

The authors should also please comment on the wound assay shown in Fig. 4G. The cells in the control siRNA + Sema4D at 24 h seem to be much more dense, which would facility filing the wound. Please comment.

Response: We thank for the reviewer's careful observation. Endothelial cell proliferation and migration may both facilitate the wound repair in vitro. Although endothelial cells were cultured in DMEM with 0.5% FBS containing low concentration of mitomycin-C to minimize the impact of cell proliferation during the wound repair, this condition can't completely avoid endothelial cell proliferation. Previous studies [6, 7] indicated that phosphorylation of Src kinase could induce endothelial cell proliferation. It is possible that Sema4D may induce endothelial cell proliferation though phosphorylation of Src kinase. So it made sense that the more cell dense in the control siRNA + Sema4D at 24 h may partly due to the stimulating effect of Sema4D on endothelial cell proliferation. However, since this study already contained a large amount of data, there was no more space for us to further explore the effect of Sema4D on endothelial cell proliferation. The most important finding of this study was that we indicated an alternative therapeutic strategy with inhibition of Sema4D/PlexinB1 to complement or improve current anti-VEGF treatment in DR.

We would like to thank you again for all comments of our manuscript. We hope that our responses have adequately addressed all points you raised.

Referee #2 (Comments on Novelty/Model System for Author):

In this revised version, the authors implement new experiments and clarify their findings. I still have a few suggestions that the authors may want to consider to further improve their manuscript.

Response: We thank for the reviewer's positive comment.

Referee #2 (Remarks for Author):

In this revised version, the authors implement new experiments and clarify their findings. I still have a few suggestions that the authors may want to consider to further improve their manuscript.

- Data presented in Fig S6 are not convincing and the exact procedure is not clear. Sema4D did not change N-cadherin level (S6C, pericyte), while it does in panel S6E (coculture?). This requires further clarification. It would have been interesting to test VE-cadherin and N-cadherin in each cell type independently. My initial comment did not mention coculture experiments.

Response: We are very grateful for the reviewer's in-depth considerations. We are sorry that we have not described clearly about these panels. For internalization and immunoprecipitation assays, the cells were treated with recombinant Sema4D or VEGF for four hours. For Western blot assays of VE-cadherin and N-cadherin, the cells were treated with recombinant Sema4D or VEGF for twenty-four hours. Previous studies [8, 9] indicated that the dissociation of Cadherins/p120-catenin/ β -catenin complex is earlier than lysosomal degradation. So it made sense that Sema4D did not change N-cadherin in pericyte immunoprecipitation assays at four hours (Fig S6C), while it did in Western blot assays of coculture at twenty-four hours (new Fig S6J). We described it clearer in the figure legend in our revised manuscript.

In addition, we tested the expression of VE-cadherin and N-cadherin in each cell type independently as suggested. We found that Sema4D and VEGF both down-regulated VE-cadherin and N-cadherin expression in endothelial cells (new Fig S6E). However, only Sema4D decreased N-cadherin in pericytes (new Fig S6G). The effect of VEGF on N-cadherin seemed to be slightly in the coculture model. We speculated that this might due to pericytes expressed much more N-cadherin than endothelial cells (new Fig S6I - 6K). These results indicated that Sema4D displayed an additional effect than VEGF on N-cadherin expression of pericytes in vitro.

- From the description of CRISPR cell lines, it seems that the authors used a pool of knockdown cells. This should be clearly indicated (no sequencing to determine mono/bi-allelic recombination?).

Response: We thank for the reviewer's careful observation. However, unlike cancer cell lines, the primary mouse microvascular endothelial cells and pericytes have a limited passage number and are difficult to be clonally expanded. So we used a pool of knockdown cells with CRISPR/Cas9 virus according to previous studies [10, 11] and did not obtain bi-allelic knockout cell lines. As suggested, we have indicated this clearer in the revised manuscript (added in the "Viral vectors" section).

- Sema3A might impact on Src activation, and VE-cadherin internalization (Le Guelte et al, J Cell Sci 2012); this putative contribution of Sema3A to the observed phenotype should not be underestimated, as Sema3A may be part of the mechanism.

Response: We fully agree with the reviewer that Sema3A may contribute to the observed phenotype. We discussed more in the revised manuscript.

Referee #3 (Remarks for Author):

In this revised paper, authors have adequately addressed my previous criticism and strengthened the data. In particular the data in Figure 6 are intriguing and convincing. Now the paper is acceptable.

Response: We sincerely thank the reviewer's previous comments. Your comments are very helpful for us to improve our work.

1. Shan, K., et al., *Circular Noncoding RNA HIPK3 Mediates Retinal Vascular Dysfunction in Diabetes Mellitus*. *Circulation*, 2017. **136**(17): p. 1629-1642.
2. Liu, C., et al., *Silencing Of Circular RNA-ZNF609 Ameliorates Vascular Endothelial Dysfunction*. *Theranostics*, 2017. **7**(11): p. 2863-2877.
3. Liu, C., et al., *Targeting pericyte-endothelial cell crosstalk by circular RNA-cPWWP2A inhibition aggravates diabetes-induced microvascular dysfunction*. *Proc Natl Acad Sci U S A*, 2019. **116**(15): p. 7455-7464.

4. Hu, J., et al., *Inhibition of soluble epoxide hydrolase prevents diabetic retinopathy*. Nature, 2017. **552**(7684): p. 248-252.
5. Yan, B., et al., *lncRNA-MIAT regulates microvascular dysfunction by functioning as a competing endogenous RNA*. Circ Res, 2015. **116**(7): p. 1143-56.
6. Kim, M.P., et al., *Src family kinases as mediators of endothelial permeability: effects on inflammation and metastasis*. Cell Tissue Res, 2009. **335**(1): p. 249-59.
7. Toutounchian, J.J., et al., *Novel Small Molecule JP-153 Targets the Src-FAK-Paxillin Signaling Complex to Inhibit VEGF-Induced Retinal Angiogenesis*. Mol Pharmacol, 2017. **91**(1): p. 1-13.
8. Mukherjee, S., M. Tessema, and A. Wandinger-Ness, *Vesicular trafficking of tyrosine kinase receptors and associated proteins in the regulation of signaling and vascular function*. Circ Res, 2006. **98**(6): p. 743-56.
9. Gavard, J., *Endothelial permeability and VE-cadherin: a wacky comradeship*. Cell Adh Migr, 2013. **7**(6): p. 455-61.
10. Gao, Y., et al., *Loss of ERalpha induces amoeboid-like migration of breast cancer cells by downregulating vinculin*. Nat Commun, 2017. **8**: p. 14483.
11. Gong, H., et al., *Method for Dual Viral Vector Mediated CRISPR-Cas9 Gene Disruption in Primary Human Endothelial Cells*. Sci Rep, 2017. **7**: p. 42127.

Corresponding Author Name: Bo Hu
Journal Submitted to: EMBO Molecular Medicine
Manuscript Number: EMM-2018-10154